# UPF1 promotes the formation of R loops to stimulate DNA double-strand break repair

Greg H. P. Ngo [1], Julia W. Grimstead[1] & Duncan M. Baird [1]✉

DNA-RNA hybrid structures have been detected at the vicinity of DNA double-strand breaks (DSBs) occurring within transcriptional active regions of the genome. The induction of DNA-RNA hybrids strongly affects the repair of these DSBs, but the nature of these structures and how they are formed remain poorly understood. Here we provide evidence that R loops, three-stranded structures containing DNA-RNA hybrids and the displaced single-stranded DNA (ssDNA) can form at sub-telomeric DSBs. These R loops are generated independently of DNA resection but are induced alongside two-stranded DNA-RNA hybrids that form on ssDNA generated by DNA resection. We further identified UPF1, an RNA/DNA helicase, as a crucial factor that drives the formation of these R loops and DNA-RNA hybrids to stimulate DNA resection, homologous recombination, microhomology-mediated end joining and DNA damage checkpoint activation. Our data show that R loops and DNA-RNA hybrids are actively generated at DSBs to facilitate DNA repair.

[1] Division of Cancer and Genetics, School of Medicine, Cardiff University, Cardiff, UK. ✉email: bairddm@cardiff.ac.uk

Telomeres are nucleoprotein structures at the end of linear chromosomes that protect the DNA from being recognized as DNA double-strand breaks (DSBs). However, telomeres become progressively shortened with each cell division in somatic cells due to low telomerase activity, which eventually leads to telomere dysfunction and the activation of the DNA damage response[1]. Dysfunctional telomeres are subjected to fusion by classical-non homologous end-joining (C-NHEJ) driven by ligase 4 (LIG4) or alternative-non homologous end-joining (A-NHEJ) driven by ligase 3 (LIG3) or ligase 1 (LIG1)[2–4]. A curious feature observed at fused telomeres is the presence of extensive deletion at fusion junctions, that can extend many kilobases into the telomere-adjacent DNA[3,5,6]. The association of deletion events with microhomology (MH) suggests that A-NHEJ might be involved in mediating telomere fusion[6], however, the mechanisms that stimulate such extensive deletion remain poorly understood.

R loops are generated when single-stranded RNA anneals to one strand of DNA, forming three-stranded structures containing DNA-RNA hybrids and the displaced non-template single-stranded DNA (ssDNA)[7]. Extending for up to several hundred bases, R loops are distinct from the transient DNA:RNA hybrids that can occur during transcription; they have been found in various regions in the genome to regulate gene expression, but by exposing ssDNA or causing transcription–replication conflicts, they also represent a potential source of genome instability[7,8]. Interestingly, recent studies have detected the presence of DNA–RNA hybrids at the vicinity of DSBs, especially those occurring within transcribed regions, and these DNA–RNA hybrids are important for the recruitment of various DNA repair proteins involved in homologous recombination (HR) or NHEJ[9–13]. However, whether these structures are DNA-RNA hybrids (two-stranded structures) or R loops (three-stranded structures with the displaced ssDNA) is unclear. These studies utilized either the S9.6 antibody or RNaseH1 to identify DNA–RNA hybrids, but both these methods cannot differentiate between DNA–RNA hybrids and R loops. The evidence for the formation of R loops in the genome have been provided indirectly from the analysis of sodium bisulfite induced mutations on the displaced ssDNA, although such an approach has not been used to analyse the DNA–RNA hybrids at DSBs[14,15]. The requirement for DNA resection to generate these structures suggests that two-stranded DNA–RNA hybrids form on ssDNA generated by DNA resection[9,11]. However, the requirement of structure specific flap endonuclease XPG in the processing of these structures at DSBs implicate the involvement of R loops[13]. In addition, it remains uncertain whether such structures are actively generated to promote repair or represent accidental structures that need to be removed before DNA repair can proceed[8].

UPF1 is an RNA/DNA helicase that was first identified as a gene required for stimulating nonsense-mediated decay (NMD), the decay of mRNAs harboring premature termination codons[16]. Further studies showed that UPF1 also regulates the degradation of normal mRNA transcripts[17]. The current model is that UPF1 recognizes certain unusual features in mRNA and recruits downstream factors to facilitate the degradation of these mRNAs in the cytoplasm[17]. However, various studies have shown that UPF1 also associates with nascent transcript inside the nucleus, but the function remain undefined[18,19]. Furthermore, UPF1 has a non-canonical and enigmatic role in the maintenance of genome stability at telomeres[20,21]. In this study, we identified UPF1 as a gene that strongly stimulates deletion at sub-telomeric DSBs. By using strand specific detection of ssDNA at DSBs coupled with detection of DNA–RNA hybrids, we discovered that sub-telomeric DSBs induced both DNA–RNA hybrids and R loops, and that UPF1 is an important factor that promotes the formation of these structures to stimulate DNA resection and repair of DSBs.

## Results

**DNA resection contributes to deletion formation at telomere fusion.** To understand the mechanism of telomere fusion, we have previously designed Transcription Activator-Like Effector Nucleases (TALEN) that cleave the sub-telomeric regions of two families of related telomeres that include 21q and 16p, between them encompassing at least 19 chromosome ends (Supplementary Fig. 1a). The 21q/16p TALEN cleaves the 21q and 16p family telomeres at a site which is located at 1494 bp (21q) or 1027 bp (16p) from the start of the telomere repeat arrays[4]. We detected telomere fusions that arise following nucleolytic cleavage using a single-molecule PCR based telomere fusion assay that targets the 16p and 21q families of telomeres[4,6]. Primer 16p1 recognizes a sequence located at 5058 bp proximal to the cleavage site on the 16p telomeres, whereas primer 21q1 recognizes a sequence located at either 2176 bp or 1856 bp proximal from the cleavage site (depending on whether the 21q family member contains a variant 320 bp deletion; Fig. 1a). We transiently expressed the 21q/16p TALEN by nucleofection into RPE1-hTERT (RPE1) cells before harvesting the cells after 48 h. Fusion PCR using both 21q1 and 16p1 primers generated two strongly hybridizing bands, detectable with both the 21q or the 16p specific probes, with sizes that were consistent with unprocessed fusion between the 21q and 16p telomeres (6.9 and 7.2 kb, Fig. 1a, b and Supplementary Fig. 1b) and this was confirmed by Sanger sequencing of the purified fusion products (Fig. 1c). The fused 16p telomere was deleted by 59 bp whereas the fused 21q telomere was deleted by 2 bp and there was no microhomology at the fusion junction, implicating classical-non homologous end joining (C-NHEJ) in the fusion (Fig. 1c). To understand the kinetics of the fusion process, we performed time course experiments (Fig. 1d and Supplementary Fig. 1c). Surprisingly, we found that these unprocessed molecules could be detected as early as 4 h post nucleofection and their level steadily increased up to 10 h and remained stable up to 96 h (Fig. 1d and Supplementary Fig. 1c). We also found that many fusion molecules larger than the unprocessed fusion (insertions) started to accumulate after 24 h, whereas many fusion molecules smaller than the unprocessed fusion (deletions) started to appear after 48 h (Supplementary Fig. 1c). Overall, these results confirm that the 21q/16p TALEN cleaves 21q and 16p family telomeres efficiently to induce telomere fusion, which are either minimally processed or are associated with large insertion or deletion.

To simplify the fusion assay, we performed single primer fusion PCR to detect fusion within the 21q family members only. We detected a constant band just under 4 kb, which was consistent with an unprocessed fusion between 21q family chromosomes, together with both smaller (deletions) and larger (insertions) fusion products (Fig. 1e). However, this could not be confirmed as this molecule was refractory to reamplification and sequencing, possibly due to its extreme palindromic nature. Nevertheless, we managed to sequence smaller fusion products (deletions) and confirmed that these were indeed fusions within 21q family telomeres and associated with large deletion events (Fig. 1f, g). Consistent with previous observations the fusion junctions exhibited microhomology, and non-templated insertions, together with asymmetric deletion with one fused telomere extensively deleted and the other minimally deleted (Fig. 1f, g)[3,5,6].

We utilized previously generated *LIG4* and *LIG3* knockouts in the HCT116 background to examine the roles of the C-NHEJ and A-NHEJ pathways in mediating the different classes of fusion events detected following cleavage with the 21q/16p TALENs[2,22]. Consistent with a role of C-NHEJ in generating unprocessed fusion, deletion of *LIG4* strongly reduced the frequency of these fusion products in both 21q1 and 21q + 16p1 assays to about 9% of the WT level ($p < 0.0001$), whereas *LIG3* deletion had a much

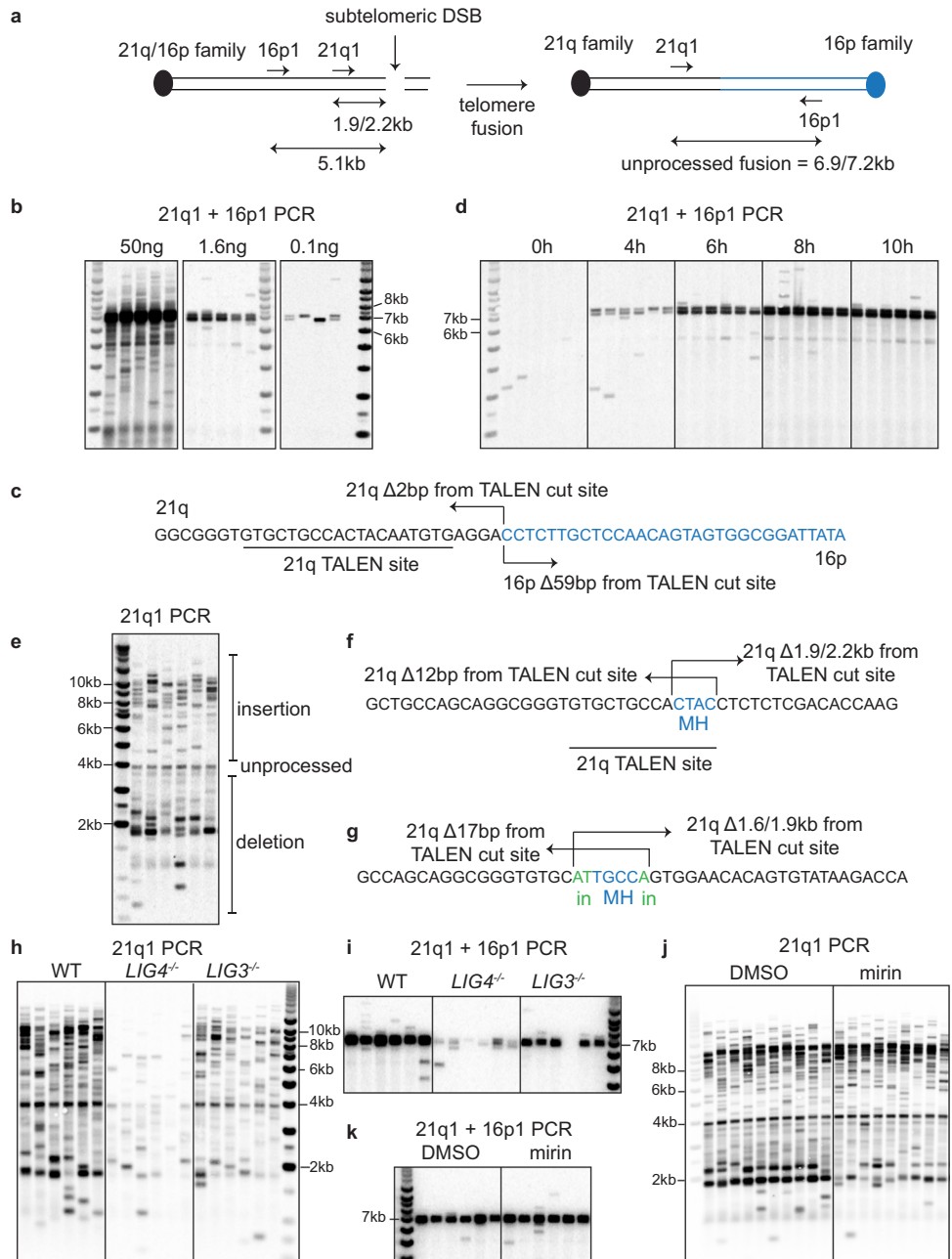

**Fig. 1 DNA resection contributes to deletion formation at telomere fusion. a** Diagram of 21q/16p TALEN induced telomere fusion PCR assay, with position of primers, the distance of primers to the cleavage site and the expected size of fusion product indicated. **b** 21q1 + 16p1 telomere fusion analysis (a representative image from two independent experiments) using the indicated amount of DNA isolated from RPE1-hTERT (RPE1) cells 48 h after TALEN nucleofection. Telomere fusion products were detected with a 16p telomere adjacent probe. **c** Fusion between 21q and 16p family telomeres characterized by Sanger sequencing of re-amplified and purified PCR products, with the number of nucleotides deleted indicated by Δbp. **d** 21q1 + 16p1 telomere fusion assay using DNA (50 ng) isolated from RPE1 cells at the indicated time after nucleofection. **e** 21q1 telomere fusion assay (a representative image from four independent experiments) using DNA (50 ng) isolated from RPE1 cells 48 h after nucleofection. Telomere fusion products were detected with a 21q telomere adjacent probe. **f**, **g** Examples of fusion between 21q family telomeres characterized by Sanger sequencing of re-amplified and purified PCR products, with the number of nucleotides deleted (Δbp), microhomology (MH) and insertions (in) indicated. **h**, **i** 21q1 or 21q1 + 16p1 telomere fusion assay using DNA (50 ng in h, 1.6 ng in **i**) isolated from HCT116 WT or LIG3/LIG4 KO cells 48 h after nucleofection, with quantification of fusion bands from three independent experiments shown in Supplementary Fig. 1d–e. **j**, **k** 21q1 or 21q1 + 16p1 telomere fusion assay using DNA (50 ng in **j**, 1.6 ng in **k**) isolated from RPE1 cells treated with DMSO or 50 μM mirin 48 h after nucleofection, with quantification of fusion bands from two/three independent experiments shown in Supplementary Fig. 1f–g.

smaller reduction (64 and 52%, $p = 0.1755$ and $0.0097$) (Fig. 1h, i and Supplementary Fig. 1d, e). *LIG4* deletion also had a much stronger effect on reducing insertions than *LIG3* (4% in LIG4 KO $p < 0.0001$ vs. 51% in LIG3 KO, $p = 0.0038$), suggesting that C-NHEJ at these breaks are prone to incorporating big insertions. Surprisingly, the absence of *LIG4* also strongly reduced the formation of deletions (27%, $p = 0.0015$), to a similar extent as in the absence of *LIG3* (43%, $p = 0.0507$), suggesting that both C-NHEJ and A-NHEJ are responsible for ligating telomere fusions associated with deletion (Fig. 1h and Supplementary Fig. 1d). This finding supports the observation that C-NHEJ can also be associated with deletion and microhomology[23]. We next examined the role of DNA resection, required for both C-NHEJ and A-NHEJ[23,24], in mediating telomeric deletion prior to fusion, by treating RPE1 cells with mirin, which inactivates Mre11 and DNA resection activity[25]. Consistent with our hypothesis, mirin strongly reduced the formation of deletions to 22% of WT level but had no major effect on unprocessed fusion or insertions (Fig. 1j, k and Supplementary Fig. 1f, g). In addition, co-inhibition of Mre11 and BLM, which plays overlapping role in activating DNA resection[26], further reduced the level of deletion (Supplementary Fig. 1h, i). We conclude that at sub-telomeric DSBs, DNA resection is responsible for generating large deletions at telomeres fused by either the C-NHEJ or A-NHEJ pathways.

**UPF1 is involved in promoting telomere deletion**. To further understand the mechanisms regulating DNA resection and deletion at telomeric DSBs, we carried out an siRNA screen to examine how the silencing of a panel of DNA damage response genes affects the formation of deletions in the 21q1 assay. From this screen, UPF1 was identified as a potential modulator of deletion; knockdown of UPF1 strongly reduced the formation of deletions and insertions without affecting the unprocessed fusion (Fig. 2a, b). To confirm this finding, we used CRISPR-Cas9[D10A] nickase[27] to knockout (KO) *UPF1* in RPE1 cells and isolated two clones (46 and 54) which showed severe depletion of the UPF1 protein, with *UPF1* 54 being a more complete KO than *UPF1* 46, which is a partial KO (Fig. 2c). A previous study had shown that depleting UPF1 in HeLa caused a severe proliferation defect by arresting all cells in early S phase[20]. We found that compared to the WT cells the *UPF1* KO clones displayed slightly more cells in G1 (Fig. 2d), however whilst they grew approximately 45% slower (Supplementary Fig. 2a), these cells could still be propagated in long-term culture, suggesting that the terminal phenotype previously observed could be cell type specific[20]. It has also been shown that depleting UPF1 leads to loss of telomeres[21], however single telomere length analysis (STELA) showed that these *UPF1* KO clones displayed slightly longer telomere-length distributions compared to WT ($p < 0.0001$ for *UPF1* 54 and $p = 0.0006$ for *UPF1* 46, Mann–Whitney *T*-test, Fig. 2e). To confirm the loss of UPF1 function in these KO clones, we examined two well characterized markers of NMD, *GADD45B*, and *NAT9*[28,29]. The mRNA level of both *GADD45B* and *NAT9* were strongly increased in *UPF1* KO clones compared to WT, confirming a loss of NMD activity (Fig. 2f). Furthermore, this phenotype could be partially suppressed by transient expression of UPF1 on a plasmid, thus confirming that the NMD defect is due to a loss of UPF1 (Supplementary Fig. 2b). To confirm the role of UPF1 in generating deletion at fused telomeres, we examined the fusion spectrum in *UPF1* KO cells. Importantly, we found that deletion of *UPF1* strongly reduced the formation of deletions to about 40% of the WT level (Fig. 2g, h). However, *UPF1* KO had no impact on the unprocessed fusion or insertion events (Fig. 2g–i and Supplementary Fig. 2c), suggesting that the effect of UPF1 knockdown in reducing insertions (Fig. 2a) could possibly be due

to off target effects. Alternatively, the effect on insertion could be masked by suppressor accumulation or genetic compensation in *UPF1* KO cells. The effect of *UPF1* KO on reducing deletion was not due to differences in transfection efficiency (Supplementary Fig. 2d). Intriguingly, transient expression of UPF1 did not restore, but further reduced the level of deletion, in addition to reducing insertion events in *UPF1* KO cells, with the effect being stronger in *UPF1* 46 than *UPF1* 54 (Supplementary Fig. 2e–f), suggesting that ectopic expression of UPF1, could exert a dominant negative effect on telomere processing, despite having a positive effect on NMD (Supplementary Fig. 2b). This dominant negative phenotype was also observed in WT cells (Supplementary Fig. 2g). This result suggests that the function of UPF1 at telomere repair could be distinct from its role in NMD, and that this function is highly sensitive to perturbation in the level of UPF1. We conclude that UPF1 strongly stimulates deletion formation and could also affect insertion at sub-telomeric DSBs.

**UPF1 stimulates DNA resection at sub-telomeric DSBs independently of NMD**. As DNA resection strongly stimulates telomere deletion (Fig. 1j), we hypothesized that UPF1 might promote DNA resection at sub-telomeric DSBs. To examine DNA resection, we adopted Quantitative amplification of single stranded DNA (QAOS) approach to directly quantify the level of 3′ ssDNA at these DSBs (Supplementary Fig. 3a)[30,31]. Three sets of QAOS primers were designed to accurately quantify 3′ ssDNA at three loci located at 0.5, 1.2, and 3.5 kb proximal (centromeric) to the TALEN cleavage site (Supplementary Fig. 3b, c). Using this technique, we detected an increase in the level of 3′ ssDNA at 0.5 and 1.2 kb, 5 h following nucleofection of 21q/16p TALEN into RPE1 cells, with additional 3′ ssDNA accumulating at these loci after 7.5 and 10 h post nucleofection (Fig. 3a). We detected 3′ ssDNA at the 3.5 kb locus after 7.5 h, suggesting that it takes a minimum of 2.5 h (5–7.5 h time point) for DNA resection to proceed a distance of 2.3 kb (from 1.2 to 3.5 kb locus). Thus the minimum rate of resection at this sub-telomeric DSBs is 0.92 kb/h, which is considerably slower than the resection rates estimated at telomeres (~8 kb/h) or at non-telomeric DSBs (~4 kb/h) in budding yeast[31–33]. To further confirm that the 3′ ssDNA detected with QAOS is due to DNA resection, we examined how CtIP and 53BP1 affect the formation of these ssDNA. CtIP stimulates DNA resection whereas 53BP1 inhibits it[34]. The generation of 3′ ssDNA in RPE1 cells was greatly suppressed following CtIP knockdown, whereas the knockdown of 53BP1 had the opposite effect in strongly elevating the level of 3′ ssDNA at all loci (Fig. 3b, c). These results confirm that the 3′ ssDNA detected with QAOS following TALEN cleavage are the products of DNA resection.

We next analysed whether UPF1 contributes to telomere deletion at sub-telomeric DSBs by stimulating DNA resection. In support of this hypothesis, the generation of 3′ ssDNA following 21q/16p TALEN induced DSBs was strongly reduced in *UPF1* KO cells at all loci (Fig. 3d). DNA resection activity is tightly regulated by cell cycle, with CtIP stimulated during late S or G2 following phosphorylation by cyclin dependent kinase[35]. However, a recent observation has challenged this idea as it was shown that CtIP can also stimulate DNA resection in G1 following its activation by Plk3[23]. Since *UPF1* KO increases the level of cells in G1, we considered whether this could contribute to a reduction in DNA resection in these cells (Fig. 3d). To test this, we compared RPE1 cells plated at different cell densities and thus have different level of cells in G1 before nucleofection with 21q/16p TALEN (Supplementary Fig. 3d). Interestingly, we found that a ~10% increase in the level of G1 cells did not cause a significant reduction in the level of 3′ ssDNA (Supplementary Fig. 3e),

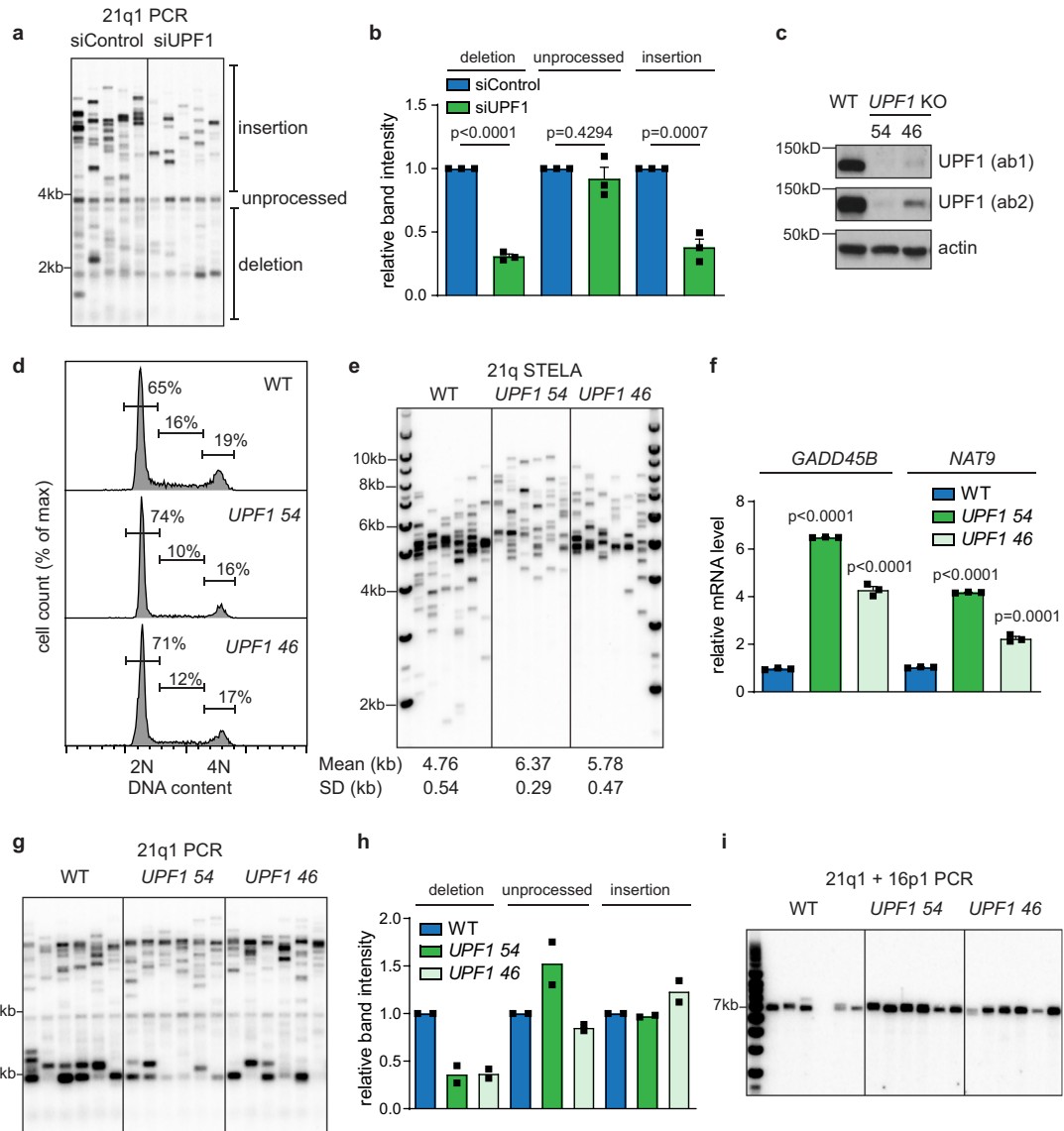

**Fig. 2 UPF1 is involved in promoting telomere deletion. a** 21q1 telomere fusion assay using DNA (50 ng) isolated from RPE1 cells transfected with UPF1 or control siRNAs 48 h after TALEN nucleofection. **b** Bar chart showing quantification of the intensity of fusion bands in siControl and siUPF1 (relative to the values in siControl). Data plotted are mean + SEM ($n = 3$). $p$ values were obtained using Student's $t$-test (unpaired two-tailed, equal variance). **c** Western blot (a representative image from two independent experiments) showing the level of UPF1 (detected using two different antibodies ab1 or ab2) and actin in RPE1 WT or *UPF1* KO cells. **d** Cell cycle analysis of RPE1 WT or *UPF1* KO cells. **e** 21q STELA of RPE1 WT and *UPF1* KO cells. A representative image from two independent STELA analysis was shown. **f** RT-qPCR analysis showing the mRNA level of two NMD markers *GADD45B* and *NAT9* in WT or *UPF1* KO cells. The values shown are relative to the values in WT and normalized with the mRNA level of *GAPDH*. Data plotted are means + SEM ($n = 3$). $p$ values were obtained using Student's $t$-test (unpaired, two-tailed, equal variance). **g** 21q1 telomere fusion analysis using DNA (50 ng) isolated from RPE1 WT or *UPF1* KO cells 48 h after nucleofection. **h** Bar chart showing quantification of the intensity of fusion bands in RPE1 WT or *UPF1* KO cells (relative to values in WT). Data plotted are means ($n = 2$). **i** 21q1 + 16p1 telomere fusion analysis using DNA (0.8 ng) isolated from RPE1 WT and *UPF1* KO cells 48 h after nucleofection, with quantification of fusion bands from three independent experiments shown in Supplementary Fig. 2c. The $n$ values indicate the number of independent experiments in all cases.

suggesting that the resection defect observed in *UPF1* KO cells is not due to a difference in cell cycle. We conclude that QAOS allows direct quantification of DNA resection at sub-telomeric DSBs and that UPF1 plays an important role in stimulating DNA resection at these loci.

The canonical function of UPF1 is to stimulate RNA degradation to promote NMD. We next investigated whether the resection defect in *UPF1* KO cells arose due to a defect in NMD (Fig. 2f). To test this hypothesis, we examined whether knockdown of UPF2 and UPF3B, two core partners of UPF1 in NMD, also affected DNA resection. Efficient knockdown of UPF1

led to a strong increase in the level of two NMD markers, *GADD45B* and *NAT9* ($p \leq 0.001$; Fig. 3e and Supplementary Fig. 3f), as did the knockdown of UPF2. But interestingly, the highly efficient knockdown of UPF3B showed no NMD defect (Fig. 3e and Supplementary Fig. 3f), which could be due to the requirement for small amounts of UPF3B for NMD or the presence of UPF3B-independent pathways[36]. We next nucleofected 21q/16p TALEN into RPE1 cells two days following the knockdown of UPF1, UPF2, or UPF3B to compare the effect of silencing these genes on DNA resection. Consistent with a defect in resection observed in *UPF1* KO cells, UPF1 knockdown

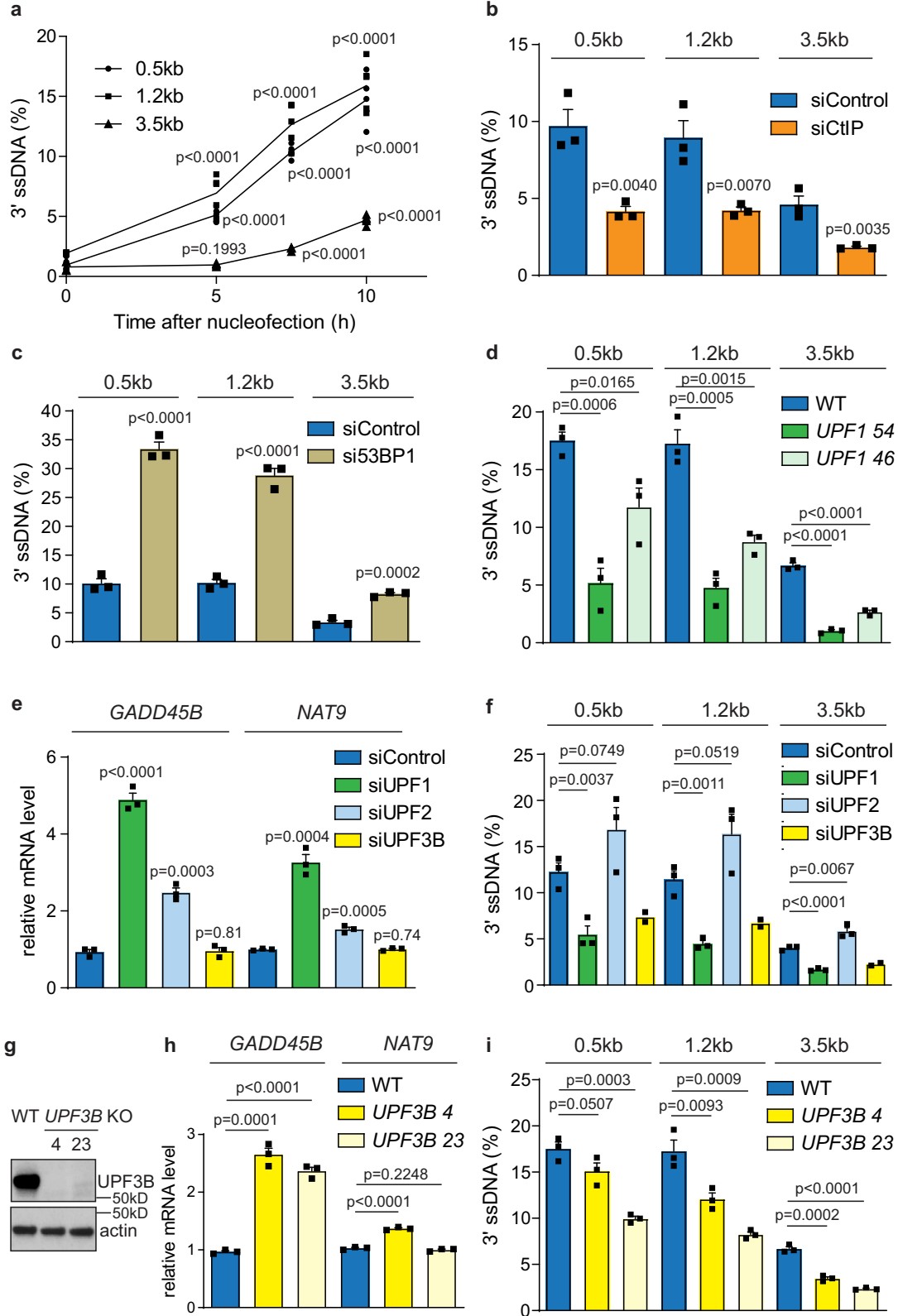

strongly reduced 3′ ssDNA at all loci (Fig. 3f). On the contrary, knocking down UPF2 did not reduce 3′ ssDNA formation, despite causing an NMD defect (Fig. 3f), suggesting that the effect of UPF1 on DNA resection is at least partially independent on NMD. The knockdown of UPF3B strongly suppressed DNA resection despite having no effect on NMD (Fig. 3f), further indicating a lack of correlation between NMD and DNA

resection, and that the role of UPF1 in DNA resection may require its interaction with UPF3B. To further probe the role of UPF3B, we created two *UPF3B* KO clones (4 and 23) using CRISPR-Cas9$^{D10A}$ nickase (Fig. 3g). Interestingly, these cells exhibited a modest NMD defect compared to *UPF1* KO cells (Figs. 2f and 3h) but nevertheless had a similar defect in stimulating DNA resection, which was comparable to the defect

**Fig. 3 UPF1 stimulates DNA resection at sub-telomeric DSBs independently of NMD. a–d** QAOS analysis showing the level of 3'ssDNA at three loci located at 0.5, 1.2, and 3.5 kb away from the DSB in RPE1 WT cells at the indicated time after TALEN nucleofection (**a**) or RPE1 cells transfected with CtIP or control siRNAs (**b**), RPE1 cells transfected with 53BP1 or control siRNAs (**c**), and RPE1 WT or *UPF1* KO cells (**d**) at 10 h after nucleofection. Data plotted are mean + SEM ($n = 5$ for **a**, $n = 3$ for **b–d**). $p$ values were obtained using Student's $t$-test (unpaired one-tailed, equal variance). **e** RT-qPCR analysis in RPE1 WT cells transfected with UPF1, UPF2, UPF3B, or control siRNAs. The values shown are relative to the values in siControl and normalized with the mRNA level of GAPDH. Data plotted are mean + SEM ($n = 3$). $p$ values were obtained as described in Fig. 2f. **f** QAOS analysis in RPE1 cells transfected with UPF1, UPF2, UPF3B or control siRNAs at 10 h after nucleofection. Data plotted are mean + SEM ($n = 3$ for siControl, siUPF1, and siUPF2, $n = 2$ for siUPF3B). $p$ values were obtained as described in **a**. **g** Western blot analysis (a representative image from two independent experiments) showing the level of UPF3B and actin in RPE1 WT and *UPF3B* KO cells. **h** RT-qPCR analysis in WT or *UPF3B* KO cells. The WT used was the same as in Fig. 2f. Data plotted are mean + SEM ($n = 3$). $p$ values were obtained as described in Fig. 2f. **i** QAOS analysis in RPE1 WT or *UPF3B* KO cells (The WT used was the same as in **d**) at 10 h after nucleofection. Data plotted are means + SEM ($n = 3$). $p$ values were obtained as described in **a**. The $n$ values indicate the number of independent experiments in all cases.

observed in *UPF1* KO 46 (Fig. 3i, d), again suggesting that the defects in DNA resection observed in *UPF1* and *UPF3B* KO is not linked to reductions in NMD. To further uncouple the roles of UPF1 in DNA resection and NMD, we examined the effect of the small molecule inhibitor NMDI14 on DNA resection. NMDI14 disrupts the interaction between UPF1 and downstream NMD factors, thus inhibiting NMD without affecting the activity of UPF1[37]. Treatment of RPE1 cells with NMDI14 caused an NMD defect without affecting DNA resection at sub-telomeric DSBs (Supplementary Fig. 3g, h), supporting the hypothesis that UPF1 does not require interaction with downstream NMD factors to stimulate DNA resection. We conclude that the role of UPF1 in stimulating DNA resection at sub-telomeric DSBs appears to be independent of its role in NMD, but nevertheless may require its interaction with some NMD factors such as UPF3B.

**UPF1 stimulates repair of non-telomeric DNA damage.** UPF1 has been shown previously to have a non-canonical, UPF2-independent role in maintaining genome stability but its role remains enigmatic[20]. To assess whether UPF1 regulates DNA repair at non-telomeric DSBs, we treated RPE1 WT and *UPF1* KO cells with bleomycin and examined the phosphorylation state of DNA damage response proteins after 90 min. Both *UPF1* KO clones had a strong defect in the phosphorylation of RPA32 and CHK1, but had a weaker effect on the phosphorylation of ATM and its downstream target CHK2 (Fig. 4a, b). This observation suggests that UPF1 may also stimulate DNA resection at non-telomeric DSBs, as resection has been shown to be required for the sustained activation of RPA32 and CHK1 to maintain the activation of the DNA damage checkpoint[38]. We next examined bleomycin induced DNA damage checkpoint maintenance in these cells. Consistent with previous findings, bleomycin induced a strong G2/M cell cycle arrest in WT cells (Fig. 4c, compare no drug and bleomycin)[39]. In the absence of UPF1, there were significantly less cells arrested at the G2/M stage, indicating a loss of DNA damage checkpoint activity (Fig. 4c, d). Concurrently, there was an increase of *UPF1* KO cells with G1 or sub-G1 DNA content, indicating that a failure to maintain DSBs induced G2/M checkpoint arrest might allow these cells to progress through mitosis resulting in cell death or arrest in G1 (Fig. 4c, d). We also observed this defect in G2/M cell cycle arrest in *UPF1* KO cells at lower concentrations of bleomycin or in the presence of etoposide (Supplementary Fig. 4a, b). We conclude that UPF1 stimulates the maintenance of DNA damage checkpoint following non-telomeric DSB induction.

To further investigate the role of UPF1 at non-telomeric DSBs, we utilized site specific I-SceI-induced DSB repair reporter assay[40]. We first examined the SA-GFP construct (Supplementary Fig. 4c), which is commonly used to measure DNA resection activity as the formation of GFP products is dependent on DNA resection of up to 2.7 kb DNA to reveal the homologous

sequences for single strand annealing (SSA)[41]. siRNA mediated knockdown of UPF1 strongly reduced SSA efficiency to 52% of WT level, supporting the role of UPF1 in stimulating DNA resection at DSBs (Fig. 4e). In further support of this hypothesis, depletion of UPF1 strongly suppressed HR (38%), which is also dependent on DNA resection (Fig. 4f)[42]. Interestingly, UPF1 knockdown also moderately reduced the efficiency of end-joining in the EJ5-GFP assay (73%), which detects both C-NHEJ and A-NHEJ activity (Fig. 4g), possibly reflecting a reduction of resection-driven C-NHEJ and A-NHEJ activities as observed in our sub-telomeric DSB induced fusion assay[23]. We also examined the role of UPF1 in C-NHEJ by using a plasmid recircularization assay (Fig. 4h), which is highly dependent on LIG4 (Supplementary Fig. 4d) and showed that UPF1 plays no role in this repair pathway (Fig. 4h). We conclude that UPF1 promotes DNA damage checkpoint maintenance and DNA resection-dependent repair at non-telomeric DSBs.

**UPF1 promotes DNA–RNA hybrids formation at sub-telomeric DSBs.** UPF1 has been shown to bind to telomeres[21]. To examine whether this binding extends to the sub-telomeric regions cut by the 21q/16p TALENs, we performed chromatin immunoprecipitation (ChIP). We first confirmed that our UPF1 antibody could specifically pull down UPF1 in immunoprecipitation (IP) experiments (Supplementary Fig. 5a). ChIP and detection with qPCR at multiple subtelomeric loci demonstrated that UPF1 binds to the sub-telomeric region of the 21q family of telomeres, in addition to 17p sub-telomeres, suggesting that UPF1 may directly influence the repair of sub-telomeric DSBs induced by 21q/16p TALENs (Fig. 5a). In support of this hypothesis, DSB induction at 21q sub-telomeres significantly increased the level of UPF1 detected with ChIP in the vicinity of these breaks at both early and late time points, but not at the uncut 17p sub-telomeres (Fig. 5b and Supplementary Fig. 5b). This observation is consistent with a previous finding that UPF1 binds to chromatin following ionizing radiation[20], and suggest that UPF1 may directly stimulate DNA resection at DSBs.

UPF1 has been shown to regulate the processing of telomeric repeat containing RNA (TERRA) at telomeres[21]. Specifically, depletion of UPF1 increased the level of telomere associated TERRA, prompting the authors to propose that UPF1 is responsible for removing TERRA from telomeres[21]. As TERRA contributes to DNA–RNA hybrid formation and DSBs-associated DNA–RNA hybrids affect DNA resection[9,43], we hypothesized that UPF1 might facilitate DNA resection by modulating the formation of DNA–RNA hybrids. To test this hypothesis, we first examined whether 21q/16p TALEN induces DNA–RNA hybrid formation by performing S9.6 DNA–RNA hybrid immunoprecipitation (DRIP) followed by qPCR[44]. In support of the observed induction of DNA–RNA hybrids at DSBs[10–13], we detected increased level of DNA–RNA hybrids on both sides of the DSBs

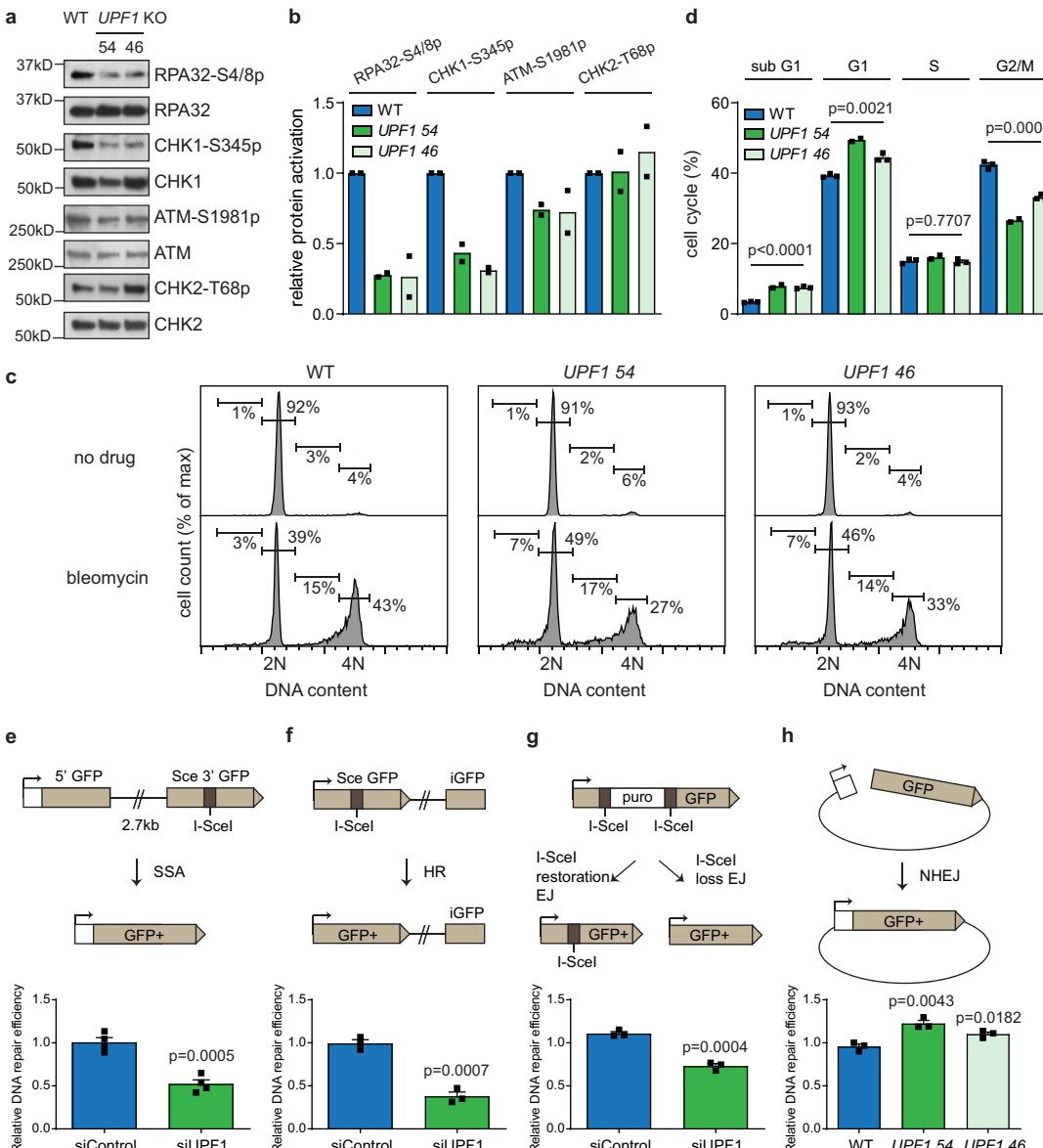

**Fig. 4 UPF1 stimulates repair of non-telomeric DNA damage. a** Western blot showing the levels of the indicated proteins in RPE1 WT or *UPF1* KO cells treated with bleomycin (10 μg/ml) for 90 min. **b** Quantification of western blot showing activation of DNA damage response proteins in RPE1 WT or *UPF1* KO cells treated with bleomycin (10 μg/ml) for 90 min. The values of phosphorylated proteins shown are relative to the values in WT and normalized with the level of total proteins. Data plotted are means (*n* = 2). **c, d** Cell cycle analysis of RPE1 WT or *UPF1* KO cells untreated or treated with 50 μg/ml bleomycin for 3 days (**c**) with the quantification of the level of sub-G1, G1, S and G2/M cells (**d**). Data plotted are mean + SEM (*n* = 3 for WT and *UPF1* 46, *n* = 2 for *UPF1* 54). *p* values were obtained using Student's *t*-test (unpaired two-tailed, equal variance). **e–g** Analysis of the efficiencies of single strand annealing (**e**), homologous recombination (**f**), and End-joining (**g**) in U2OS WT cells transfected with UPF1 or control siRNAs[40]. Data plotted are means + SEM (*n* = 4 for **e**, *n* = 3 for **f, g**). *p* values were obtained using Student's *t*-test (unpaired two-tailed, equal variance). **h** Analysis of NHEJ activity in RPE1 WT or *UPF1* KO cells. Data plotted are means + SEM (*n* = 3). *p* values were obtained using Student's *t*-test (unpaired two-tailed, equal variance). The *n* values indicate the number of independent experiments in all cases.

(0.5, 1.2, and 3.5 kb on centromeric side and 1.1 kb on telomeric side) but not at an uncut 17p or 21p telomere (Fig. 5c). Consistent with these signals being DNA–RNA hybrids, transient expression of RNaseH1 or in vitro digest with RNaseH1, which specifically degrades the RNA within DNA–RNA hybrids, strongly reduced these signals (Fig. 5d and Supplementary Fig. 5c). Next, we nucleofected the 21q/16p TALENs into RPE1 WT and *UPF1* KO cells and examined the formation of DNA–RNA hybrids by DRIP-qPCR. Surprisingly, we found that *UPF1* KO cells had lower level of DNA–RNA hybrids at all loci surrounding the breaks following DSB formation (Fig. 5e and Supplementary

Fig. 5d), which suggests that instead of removing RNA molecules such as TERRA, UPF1 might stimulate the formation of DNA–RNA hybrids at DSBs. We further hypothesized that a failure to stimulate these DNA–RNA hybrids causes the DNA resection defect in the absence of UPF1. In support of this hypothesis, transient expression of RNaseH1, which reduced DNA–RNA hybrids, was found to strongly reduce DNA resection (Fig. 5f). We conclude that sub-telomeric DSBs induced the formation of DNA–RNA hybrids, and that UPF1 recruitment stimulates the formation of DNA–RNA hybrids at DSBs to promote DNA resection.

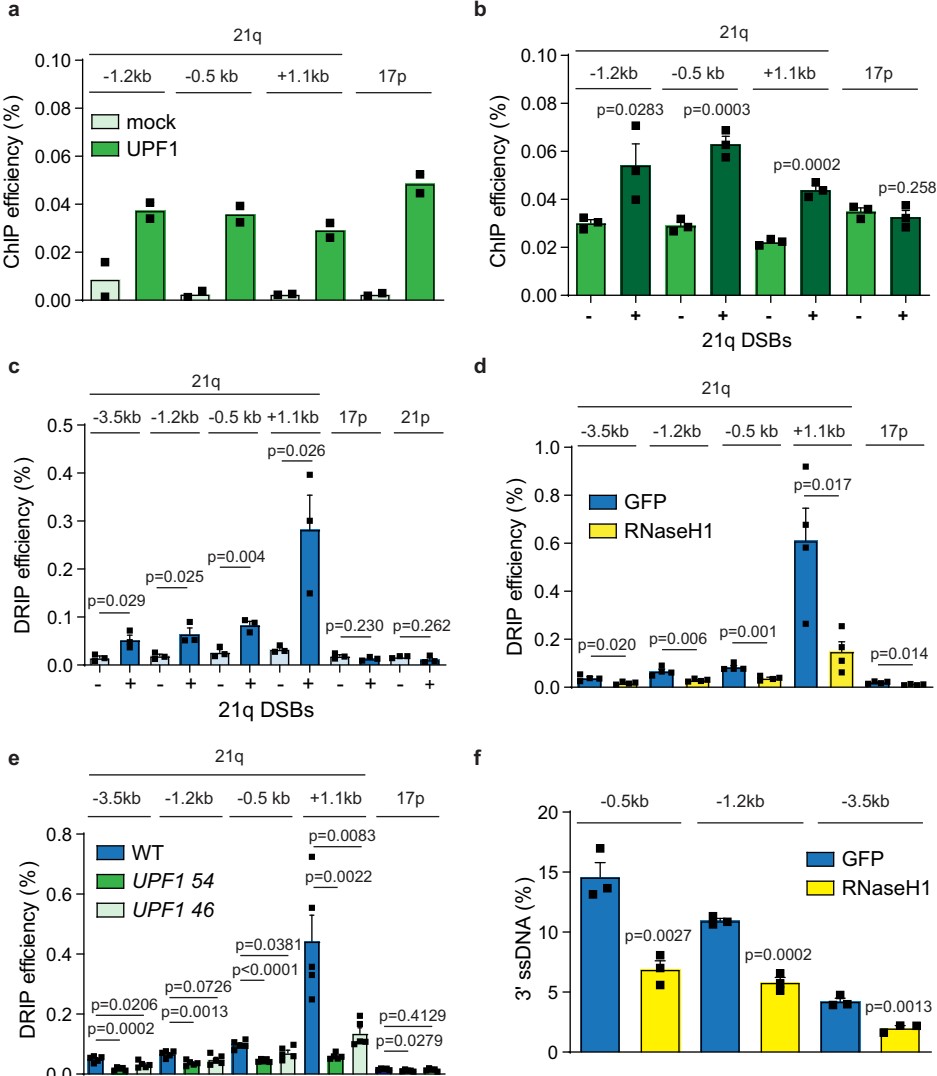

**Fig. 5 UPF1 promotes DNA-RNA hybrid formation at sub-telomeric DSBs. a, b** ChIP analysis of UPF1 binding to sub-telomeres with or without DSB induction (at 8 h after TALEN nucleofection). Data plotted are mean + SEM ($n = 2$ for **a**, $n = 3$ for **b**). $p$ values were obtained using Student's $t$-test (unpaired one-tailed, equal variance). **c–e** DRIP analysis showing the levels of DNA–RNA hybrids at the indicated loci in RPE1 WT cells in the presence (+) or absence (−) of sub-telomeric DSBs induction (**c**), RPE1 WT cells expressing RNaseH1 or GFP control (**d**), and RPE1 WT or *UPF1* KO cells (**e**) at 10 h after TALEN nucleofection. Data plotted are mean + SEM ($n = 3$ for **c, d** $n = 5$ for **e**). $p$ values were obtained using Student's $t$-test (unpaired two-tailed, equal variance). **f** QAOS analysis in RPE1 WT cells expressing RNaseH1 or GFP control at 10 h after TALEN nucleofection. Data plotted are mean + SEM ($n = 3$). $p$ values were obtained as described in Fig. 3a. The $n$ values indicate the number of independent experiments in all cases.

**UPF1 stimulates the formation of R loops at sub-telomeric DSBs.** Recent studies showed that DSB induction could induce the formation of R loops[13,45]. However, the presence of these R loops was largely inferred based on the requirement of structure-specific nuclease XPG to cleave the structure[13], or the ability of R loops to act as promoters to initiate antisense transcription[45]. To directly detect R loop structures, we developed the QAOS assay to detect 5′ ssDNA at 21q/16p TALENs induced DSBs (Supplementary Fig. 6a), which would be expected to accumulate due to R loop formation. Nucleofection of 21q/16p TALEN into RPE1 cells led to an increase in 5′ ssDNA at the 0.5 kb and 1.2 kb loci, 5 h post nucleofection, and the levels remained stable for up to 10 h at these loci, but 5′ ssDNA was not detected at the 3.5 kb locus even after 10 h (Fig. 6a). To further confirm the occurrence of these 5′ ssDNA, we developed QAOS to detect ssDNA formation on the other (telomeric) side of DSBs (Supplementary Fig. 6b). Similar to the centromeric side (Fig. 3a), DSB induction stimulated the gradual accumulation of 3′ ssDNA at the 1.1 kb

locus on the telomeric side of DSBs (Fig. 6b), which was reduced following CtIP depletion (Supplementary Fig. 6c), showing that the telomeric side was similarly subjected to DNA resection. Expression of RNaseH1 or depletion of UPF1 also reduced the accumulation of these 3′ ssDNA (Supplementary Fig. 6d and e), further supporting the role of UPF1 and DNA-RNA hybrids in stimulating DNA resection. Interestingly, we could also detect increased level of 5′ ssDNA at this locus following DSB induction (Fig. 6b), showing that 5′ ssDNA accumulates on both sides of sub-telomeric DSBs. Importantly, expression of RNaseH1 reduced the level of these 5′ ssDNA indicating that these are non-template DNA displaced by DNA-RNA hybrids and that sub-telomeric DSBs stimulates the formation of R loops (Fig. 6c). Recent studies have shown that DNA-RNA hybrids form on 3′ ssDNA created by DNA resection, which would preclude the formation of R loops[9,11]. To test this model, we examined whether the inhibition of DNA resection by depleting CtIP would affect the generation of DNA–RNA hybrids and R loops

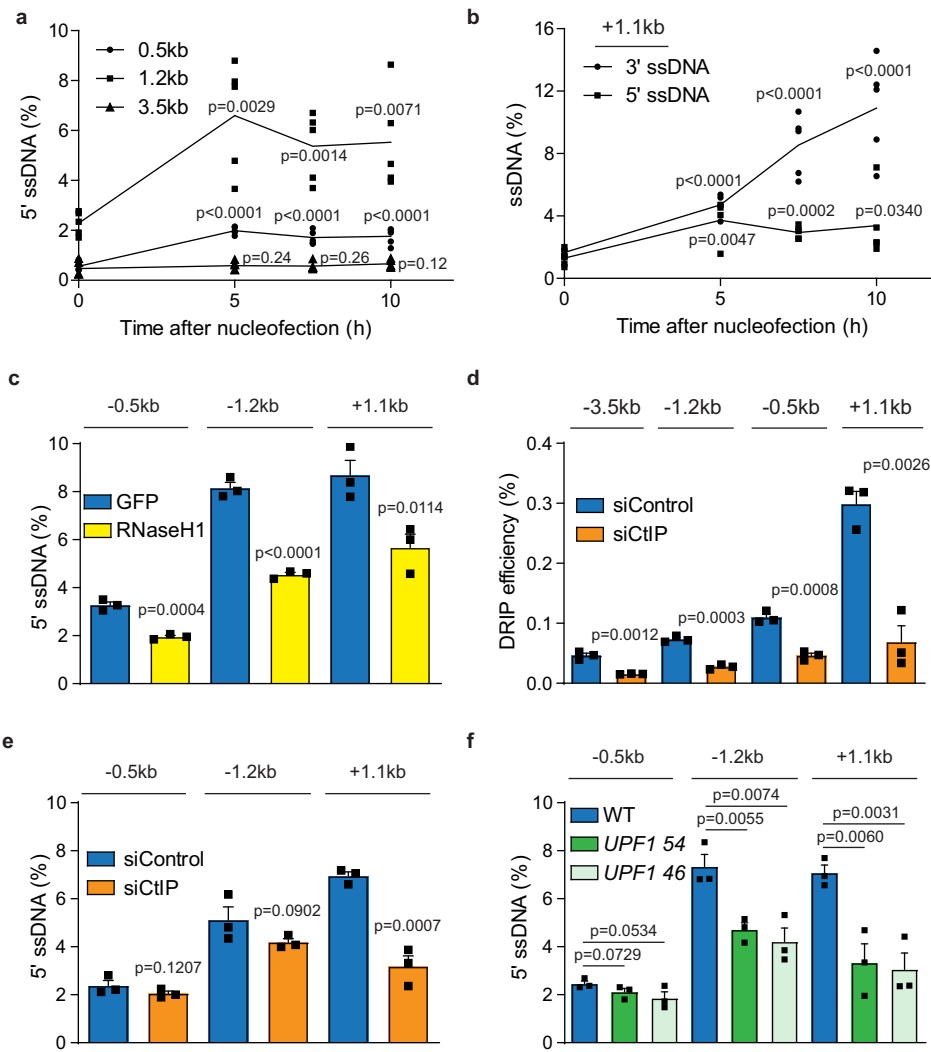

**Fig. 6 UPF1 stimulates R loop formation at sub-telomeric DSBs. a–c** QAOS analysis showing the level of 3′ and 5′ ssDNA at the indicated loci in RPE1 WT cells at the indicated time after TALEN nucleofection (**a**, **b**) or RPE1 cells expressing RNaseH1 or GFP at 10 h after nucleofection (**c**). Data plotted are mean + SEM, $n = 5$ for **a**, $n = 5$ for **b** (except for 5 h, $n = 4$), $n = 3$ for **c**. $p$ values were obtained as described in Fig. 3a. **d** DRIP analysis showing the levels of DNA–RNA hybrids at the indicted loci in RPE1 cells transfected with CtIP or control siRNAs at 10 h after nucleofection. Data plotted are mean + SEM ($n = 3$). $p$ values were obtained as described in Fig. 5c. **e**, **f** QAOS analysis in RPE1 cells transfected with CtIP or control siRNAs (**e**) and RPE1 WT and *UPF1* KO cells (**f**) at 10 h after nucleofection. Data plotted are mean + SEM ($n = 3$). $p$ values were obtained as described in Fig. 3a. The $n$ values indicate the number of independent experiments in all cases.

(5′ ssDNA) at sub-telomeric DSBs. Depletion of CtIP strongly reduced the formation of DNA–RNA hybrid, supporting the hypothesis that some DNA–RNA hybrids indeed formed on resected DNA and that some DNA–RNA hybrids detected at all loci are two stranded DNA–RNA hybrids structures (Fig. 6d). Surprisingly, CtIP depletion had mixed effects on 5′ ssDNA, by not significantly affecting the level of 5′ ssDNA on the centromeric side, but reducing 5′ ssDNA on the telomeric side, despite strongly reducing 3′ ssDNA on both sides of the break (Figs. 3b and 6e and Supplementary Fig. 6c). These results showed that some 5′ ssDNA at the centromeric side are generated independently of DNA resection, as expected for R loops structures, and that a proportion of the DNA–RNA hybrids detected at the centromeric 0.5 and 1.2 kb loci are R loop structures (Fig. 6e). However, some 5′ ssDNA at the telomeric side appear to depend on CtIP, suggesting that CtIP could be recruited to these R loops, and the cleavage of these structures by CtIP may facilitate its detection, as suggested recently (see "Discussion" section)[46]. Finally, we investigated whether UPF1 is required for the

formation of R loops. Interestingly, *UPF1* KO cells had lower level of 5′ ssDNA on both sides of DSBs, suggesting that UPF1 may also promote R loop formation (Fig. 6f). We conclude that sub-telomeric DSBs lead to the formation of both DNA–RNA hybrids and R loops, and the induction of both these structures are stimulated by UPF1.

## Discussion

Utilizing strand specific amplification of ssDNA, we have identified a novel species of DNA repair intermediates: 5′ ssDNA, at the vicinity of sub-telomeric DSBs. The sensitivity of these 5′ ssDNA to RNaseH1 suggests that these are non-template ssDNA displaced by RNA in an R loop structure, showing directly for the first time that three-stranded R loops are generated at DSBs (Fig. 7, pathway 1). We further showed that the R loop structures on the centromeric side of these DSBs are independent of DNA resection by CtIP, but can be generated alongside two-stranded DNA–RNA hybrid structures that form on CtIP-dependent ssDNA (Fig. 7, pathway 2). Intriguingly, R loops on the telomeric

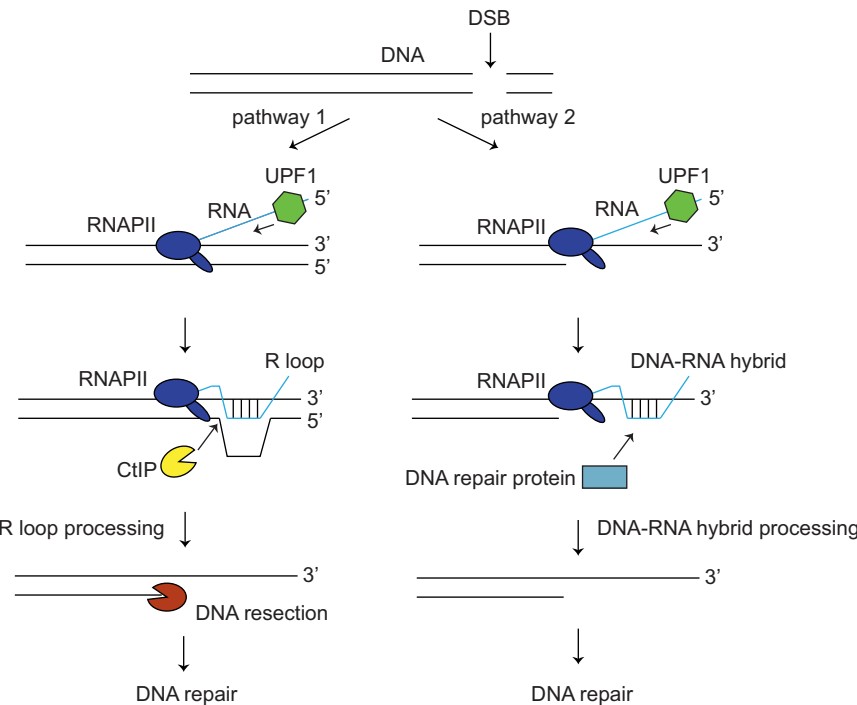

**Fig. 7 A model showing how UPF1 drives the formation of R loops and DNA-RNA hybrids to stimulate repair of DSBs.** DSBs induce the formation of R loops and DNA-RNA hybrids in two pathways. In pathway 1, UPF1 stimulates the invasion of RNA into dsDNA to form R loops. R loops recruits R loop processing factor CtIP to initiate DNA resection and resection-dependent repair. In pathway 2, UPF1 stimulates the binding of RNA to ssDNA generated by DNA resection to create DNA–RNA hybrids. These structures recruit DNA–RNA hybrid interacting proteins to promote DNA repair.

side of these DSBs appear more dependent on CtIP. We propose that these R loops are conformationally constrained, perhaps due to its proximity to the telomeric chromatin, and the detection of the 5′ ssDNA within these R loops may require nicking of the non-template ssDNA by CtIP (Fig. 7). This hypothesis is supported by the observation that CtIP, together with XPG, facilitate the detection of R loops at stalled transcription site by cleaving these structures[46]. We noted that 5′ ssDNA was not detected at the 3.5 kb locus, suggesting that R loops do not extend up to 3.5 kb from DSBs. Interestingly, DNA–RNA hybrids are detected at the 0.5, 1.2, and the 3.5 kb loci, showing that the formation of double-stranded DNA–RNA hybrid structures are more extensive than R loop structures.

Despite observations from many labs that DNA–RNA hybrids accumulate at DSBs, it is unclear how these structures are generated. It is believed that a reduction of torsional stress on the DNA double helix following DSB formation might facilitate the invasion of RNA molecules into dsDNA to form R loops[8]. Alternatively, RNA could simply hybridize with complementary ssDNA generated by DNA resection to generate DNA–RNA hybrids[9,11]. The ribonuclease Drosha has also been implicated in the formation of DNA–RNA hybrids at DSBs but the mechanism remains unclear[12]. Here, we discovered that the RNA helicase UPF1 is required for the generation of both R loop and DNA–RNA hybrid structures at sub-telomeric DSBs. We found that this role of UPF1 is independent of its canonical role in RNA decay and that UPF1 is directly recruited to DSBs to carry out this function (Fig. 7). In support of this hypothesis, UPF1 was identified as an interactor of DNA–RNA hybrids in two interactome studies, further supporting a direct role of UPF1 in DNA–RNA hybrid metabolism[47,48]. Based on the activity of UPF1 as a 5′–3′ RNA helicase/translocase which disassembles messenger ribonucleoprotein complex[49,50], we propose that UPF1 could bind to the 5′ end of RNA transcripts near DSBs (Fig. 7), and translocate toward the 3′ end of RNA to remove any secondary structures or

R loop suppressing proteins on the RNA to facilitate the annealing of these RNAs to ssDNA or dsDNA. This function may be akin to the roles of DHX9 or DDX1 in stimulating the formation of R loop structures for the regulation of gene expression or class switch recombination[51,52]. Interestingly, UPF1 also possesses the ability to translocate on DNA with a 5′–3′ polarity, unwinding dsDNA and removing any protein blocks on its way[53]. Thus, it is also possible that UPF1 could facilitate the binding of RNA to dsDNA or ssDNA by unwinding the DNA double helix or by removing any proteins on the 3′ ssDNA tail at DSBs. It would be interesting to examine these possibilities in future experiments. We found that UPF3B, an interacting partner of UPF1, is also required for DNA resection, and UPF3B has also been identified as an interactor of DNA–RNA hybrids[48]. It would be useful to test whether UPF3B regulates R loop formation by UPF1, as it has been shown to stimulate the helicase activity of UPF1[54].

Our identification of both R loop and DNA–RNA hybrid structures at DSBs could potentially help resolve the conflicting data in the literature on the effect of DNA–RNA hybrids on DNA resection. It has been shown that DSBs-induced DNA–RNA hybrids stimulate[12,13,55], exert no effect on[10,11], or hinder DNA resection[9]. Our study reveals that depletion of UPF1 or expression of RNaseH1, which reduces both R loops and DNA–RNA hybrid structures, strongly reduces DNA resection at sub-telomeric DSBs. We propose that this stimulatory effect on DNA resection is due to the induction of R loops at these DSBs before DNA are resected (Fig. 7, pathway 1). The presence of R loop structures at DSBs has been inferred previously by the requirement of structure specific flap endonuclease XPG to process such structures[13]. XPG and another endonuclease XPF, has also been implicated in cleaving R loops at sites of stalled transcription[56]. Another study found that CtIP utilizes its flap endonuclease activity to cleave R loops induced by transcription stalling[46]. Thus, we propose that R loops promote resection at these DSBs by stimulating the recruitment of XPG, XPF, or CtIP. Our

observation that CtIP is involved in processing these R loops support this hypothesis. R loop nicking by CtIP could lead to the degradation of the non-template DNA by the MRN complex, thus initiating DNA resection (Fig. 7, pathway 1). Thus, we propose that removing DNA-RNA hybrids does not reduce DNA resection in some cases likely because R loops are not generated at these DSBs[9–11]. Indeed, DNA–RNA hybrids detected in some of these experiments were shown to be totally dependent on DNA resection, supporting the hypothesis that these are double-stranded DNA–RNA hybrids which stimulate homologous recombination without affecting DNA resection (Fig. 7, pathway 2)[11]. Thus, it is important to differentiate whether the DNA–RNA hybrids detected at DSBs are double-stranded DNA–RNA hybrids or three-stranded R loop structures.

Consistent with the proposed role of R loops and DNA-RNA hybrids in stimulating DNA repair via resection-dependent or independent pathways (pathway 1 and 2, Fig. 7)[10–13], we found that UPF1 is required for homologous recombination, single strand annealing and RPA phosphorylation-induced DNA damage checkpoint activation. Interestingly, our telomere fusion assay also reveals an important role for UPF1 in stimulating resection-dependent MMEJ events at sub-telomeric DSBs, which result in telomere deletion, suggesting that sub-telomeric DSBs could be subjected to end-joining following R loop-induced resection. We note that our telomere fusion assay specifically detects NHEJ events and we cannot rule out the possibility that these R loops/DNA–RNA hybrids also stimulate homologous recombination at these sub-telomeric DSBs. We propose that UPF1 is especially important for the repair of DSBs occurring within transcriptionally active genomic regions.

In summary, we showed that both R loop and DNA–RNA hybrid structures are actively generated at the vicinity of sub-telomeric DSBs. UPF1 promotes the formation of these structures to facilitate DNA resection, homologous recombination, MMEJ, and DNA damage checkpoint activation. Our data provide novel insights into the role of RNA during the repair of DSBs and raises important questions about what factors determine whether R loops are generated at DSBs, and how such structures are processed to initiate DNA resection and repair.

## Methods

**Cell culture and analysis**. RPE1-hTERT human retinal pigment epithelial cell lines were cultured in DMEM/F12 medium (11504436, Fisher Scientific) supplemented with 1% penicillin/streptomycin, 10% FCS and 2 mM L-glutamine. HCT116 human colorectal carcinoma cell lines were grown in McCoy's 5A medium (26600080, Thermo Fisher Scientific) supplemented with 1% penicillin/streptomycin, 10% FCS and 2 mM L-glutamine. U2OS human bone osteosarcoma cell lines were grown in DMEM medium (41965062, Thermo Fisher Scientific) supplemented with 1% penicillin/streptomycin, 10% FCS and 2 mM L-glutamine. Mirin (M9948), ML216 (SML0661-5MG), and NMDI14 (SML1538) was purchased from Sigma. Cell counting were performed using a NucleoCounter NC-3000™ system (Chemometec). Cell cycle analyses were performed according to a two-step cell cycle protocol on a NucleoCounter NC-3000™ system (Chemometec).

**Plasmids, siRNA, and transfection**. Plasmids used were: 21q/16p TALENs[4], AIO-GFP (a gift from Steve Jackson, Addgene plasmid #74119; RRID:Addgene_74119), pFRT-TODestGFP_RNAseH1 (a gift from Thomas Tuschl, Addgene plasmid #65784; RRID:Addgene_65784), RNT1-GFP (a gift from Hal Dietz, Addgene plasmid #17708; RRID:Addgene_17708), pmaxGFP (Lonza), pCBASceI (a gift from Maria Jasin, Addgene plasmid #26477; RRID:Addgene_26477), pCMV-GFP (a gift from Connie Cepko, Addgene plasmid #11153; RRID:Addgene_11153) and mCherry2-C1 (a gift from Michael Davidson, Addgene plasmid #54563; RRID: Addgene_54563). siRNA used were: siUPF1, siUPF2, siUPF3B (all siGENOME SMART pool siRNA from Dharmacon), siControl: AAUUCUCCGAACGUGUCA CGUdTdT[34], siUPF1: GAUGCAGUUCCGCUCCAUUdTdT[29], siCtIP: GCUAAA ACAGGAACGAAUCdTdT[34], and si53BP1: GGACUCCAGUGUUGUCAACUdT dT[34]. Nucleofection were performed using 4D nucleofector X unit (Lonza) and SE cell line kit (program DS150 for RPE1 and DS138 for HCT116). Transfection of plasmid into RPE1 and U2OS were done using Dharmafect kb (Dharmacon) according to manufacturer's protocol. siRNA transfections were done using Dharmafect 4 (Dharmacon) for RPE1 or Lipofectamine RNAiMAX (Thermo Fisher Scientific) for U2OS according to manufacturer's protocol.

**siRNA screen**. $1.9 \times 10^5$ RPE1 cells were plated into 12 well plates on day 1. The cells were transfected on day 2 with 50 nM siRNA (Dharmacon siGENOME® SMARTpool® siRNA Library targeting Human DNA Damage Response, G-006005 Lot 14019, divided into 12 sets) using 2.5 µl Dharmafect 4 (Dharmacon) according to manufacturer's protocol. $4 \times 10^5$ siRNA transfected cells were passed onto 6 well plates on day 3 and transfected with 21q TALEN plasmids on day 4 using 3.75 µl Dharmafect kb (Dharmacon) according to manufacturer's protocol. Cells were harvested at day 7 and DNA were purified before subjected to fusion analysis.

**Creation of knock-out cell lines**. sgRNA pairs for CRISPR nickase targeting were designed using online tool CRISPR Finder provided by Wellcome Sanger Institute Genome Editing website (https://www.sanger.ac.uk/htgt/wge/). For *UPF1*, the two sites targeted were CCTGCAGAACGGGGCTGTGGACG (ch19: 18845997–18846019) and AGCCAAGACCAGCCAGTTGTTCG (ch19:18846027–18846049) on exon 2 of the gene. For *UPF3B*, the two sites targeted were CCTCGTATCATTAGAAAAAAACT (chX:119851766-119851788) and AAATAATCATGCTCAGGCATAGG (chX:119851791–119851813) on exon 2 of the gene. DNA oligos containing these sites were annealed and cloned sequentially into BbsI and BsaI sites within AIO-GFP plasmid before verification by Sanger sequencing. The plasmids carrying correct inserts were transfected into RPE1 cells using Dharmafect kb (Dharmacon). Two days later, transfected cells were sorted using BD FACSAria™ III cell sorter (BD Biosciences) and GFP positive cells were plated into 10 cm dishes at various cell density. Single colonies were isolated after about 2 weeks and positive knockout clones were identified by western blot.

**Telomere fusion PCR, reamplification, and sequencing**. Single molecule telomere fusion PCR assay were performed as described[6]. Briefly, 0.1–50 ng of phenol/chloroform extracted DNA were subjected to PCR using a 21q1 primer and/or a 16p1 primer (Supplementary Table 1). Typically, five to ten reactions containing 0.5 µM 21q1/16p1 primers and 0.5 U of a 10:1 mixture of Taq (Thermo Fisher Scientific) and Pwo polymerase (Roche) were set up and the PCR amplicons were resolved by TAE agarose gel (0.5%) electrophoresis. Fusion bands were visualized by Southern blot hybridization using a p32-labeled 21q or 16p telomere adjacent probes. Image J was used to quantify fusion band intensity. Reamplification PCR was performed as above by using fusion PCR reactions as template. Products of reamplification were resolved by TAE gel electrophoresis, and specific bands were excised, gel purified, and subjected to Sanger sequencing (Eurofins).

**Single telomere length analysis (STELA)**. STELA were performed according to standard protocols[5]. Briefly, genomic DNA was extracted using phenol/chloroform and diluted to 10 ng/µl in 10 mM Tris-HCl (pH 7.5). One microliter of this DNA solution were further diluted to 250 pg/µl in 1 mM Tris-HCl (pH 7.5) containing 1 µM Telorette2 linker. One microliter of this DNA/Telorette 2 solution were used in PCR reactions containing 0.5 µM 21q family primers 10q21T and M449 (Supplementary Table 1), 0.5 µM Teltail primer and 0.5 U of a 10:1 mixture of DreamTaq DNA polymerase (Thermo Fisher Scientific) and Pwo DNA polymerase (Roche). Following 0.5% TAE agarose gel electrophoresis and southern blotting, individual telomere bands were detected by a p32-labeled telomeric TTAGGG probe.

**Cell lysis and western blot analyses**. Cell lysis and western blot analyses were performed according to a previous study[57]. Briefly, cells were resuspended in 2.5× cell volume of lysis buffer (150 mM NaCl, 50 mM Tris HCl, 5 mM EDTA, 1% NP40, 3 mM PMSF, 1/100 protease inhibitor cocktail III [Calbiochem 539134] and 1/100 phosphatase inhibitor cocktail II [Calbiochem 524625]) and left on ice for 5 min. Following centrifugation at 20,000×g for 30 min, whole cell extracts were removed and quantified using Pierce Coomassie plus protein assay reagent (23236, Thermo Fisher Scientific). Ten to twenty microgram of protein extract were resolved by Mini-PROTEAN TGX™ precast protein gels (456-1026, Biorad) and transferred to PVDF membranes (Millipore) before probing with the following antibodies: anti-phospho-RPA32 (S4/S8) rabbit polyclonal antibody (A300-245A-M, Cambridge Biosciences, 1:500), anti-phospho-Chk1 (Ser345) rabbit monoclonal antibody (2348, Cell signaling, 1:500), anti phospho-Chk2 (Thr68) rabbit polyclonal antibody (2661, Cell signaling, 1:500), anti-UPF1 goat polyclonal antibody (A300-038A, Cambridge Biosciences, 1:40000), anti-UPF1 mouse monoclonal antibody (sc-393594, Santa Cruz, 1:120), anti-UPF3B rabbit polyclonal antibody (A303-688A, Cambridge Biosciences, 1:1000), anti-ATM(S1981) mouse antibody (200-301-400S, Cambridge Bioscience, 1:333), anti-ATM goat antibody (A300-136A-T, Cambridge Bioscience, 1:500), anti-CHK1 mouse antibody (2360S, Cell Signaling, 1:250), anti-CHK2 rabbit antibody (ab109413, Abcam, 1:3000), anti-RPA32 rabbit antibody (A300-244A-M, Cambridge Bioscience, 1:2000) and anti-actin rabbit polyclonal antibody (A2066, Sigma-Aldrich, 1:3000).

**I-SceI DNA repair assay**. SA-GFP, DR-GFP, and EJ5-GFP reporter assays were performed as previously described[40,41]. Briefly, U2OS reporter cell lines were transfected in 6-well plates on day one with siRNAs (40 nM) using lipofectamine RNAiMAX (Thermo Fisher Scientific) and transferred into 12-well plates on day two. On day three, cells were transfected using Dharmafect kb (Dharmacon) with a plasmid pCBASceI to induce I-SceI or with another plasmid mCherry2-C1 as a control for transfection efficiency. siRNAs (20 nM) were also co-transfected with

the plasmids to enhance the knockdown. On day five, cells were harvested and subjected to flow cytometry analysis using a BD LSRFortessa™ cell analyser (BD Biosciences). DNA repair efficiencies were calculated as percentage of GFP positive cells normalized with percentage of m-cherry positive cells.

**NHEJ plasmid recircularization assay.** pCMV-GFP plasmids were linearized with EcoRI and gel purified. Linearized plasmids or undigested control plasmids were nucleofected into RPE1 or HCT116 cells, and GFP positive cells were counted after 24 h according to a GFP transfection assay protocol on a NucleoCounter NC-3000™ system (Chemometec). NHEJ efficiencies were calculated as percentage of GFP positive cells due to the repair of linearized plasmid normalized with transfection efficiency of undigested plasmids.

**Quantitative amplification of ssDNA (QAOS).** The QAOS protocol were adapted from a protocol used to quantify ssDNA in budding yeast[30]. DNA were isolated using QIAamp DNA Micro Kit (56304, Qiagen) according to manufacturer's protocol with slight modifications. Briefly, cell lysis was performed at a lower temperature of 37 °C for 25 min in the presence of RNaseA (125 ng/μl) to minimize the formation of ssDNA. DNA were quantified with a QuantiFluor fluorometer (Promega) and diluted to 2 ng/μl. Ten nanogram of DNA was used in triplicate quantitative PCR reactions containing ExTaq polymerase (Takara Bio), 300 nM of forward/reverse primers, tagging primers, forward/reverse tag and 200 nM of Taqman probes. PCR were carried out using a ViiA7 Real-Time PCR machine (Thermo Fisher Scientific) and the conditions were Step 1: 40 °C for 5 min (1 cycle). Step 2: ramp to 72 °C at 2 °C / min (1 cycle). Step 3: 94 °C for 4 min (1 cycle). Step 4: 95 °C for 15 s; 67 °C for 1 min (40 cycles). For dsDNA control, the PCR conditions were Step 1: 94 °C for 5 min (1 cycle). Step 2: 95 °C for 15 s; 63 °C for 1 min (40 cycles). ssDNA standards including 10, 5, 1, 0.5, and 0.1 ng of boiled DNA were annotated as 100, 50, 10, 5, and 1% of ssDNA, and included in the PCR run to generate a standard curve. ssDNA values of DNA samples were obtained from the standard curve generated in QAOS qPCR and normalized with dsDNA values of a non-telomeric locus, obtained from normal dsDNA qPCR reactions[34]. Primers used are listed in Supplementary Table 1.

**DNA-RNA hybrid immunoprecipitation (DRIP) and qPCR.** DRIP analyses were performed as described previously with some modifications[44]. Briefly, DNA were isolated using Macherey-Nagel™ NucleoSpin™ Tissue Column (Fisher Scientific) and quantified using a QuantiFluor fluorometer (Promega). DNA were diluted to 20 ng/μl in 10 mM Tris-HCl pH8, 150 mM NaCl (200 μl final volume) and sonicated for 10 min in a Diagenode Bioruptor (4 °C, medium setting, 30 s on/30 s off). Four microliter of sonicated DNA samples were saved as input and 196 μl were subjected to DRIP overnight at 4 °C in lysis buffer (50 mM Hepes/KOH pH 7.5, 0.14 M NaCl, 5 mM EDTA pH 8, 1% Triton X-100, and 0.1% Na-Deoxycholate) containing 5 μl S9.6 antibody (MABE1095, Millipore) and 25 μl Protein G dynabead solution (10004D, Thermo Fisher Scientific). Dynabeads were isolated using magnet and washed successively (two times each) with 1 ml low salt lysis buffer (50 mM Hepes/KOH pH 7.5, 0.14 M NaCl, 5 mM EDTA pH 8, 1% Triton X-100 and 0.1% Na-Deoxycholate), 1 ml high salt lysis buffer (50 mM Hepes/KOH pH 7.5, 0.5 M NaCl, 5 mM EDTA pH 8, 1% Triton X-100 and 0.1% Na-Deoxycholate), 1 ml wash buffer (10 mM Tris-HCl pH 8, 0.25 M LiCl, 1 mM EDTA pH 8, 0.5% NP-40 and 0.5% Na-Deoxycholate), and 1 ml TE (100 mM Tris-HCl pH 8, 10 mM EDTA pH 8), before being eluted in 100 μl of elution buffer (50 mM Tris-HCl pH 8, 10 mM EDTA and 1% SDS) at 65 °C for 15 min. Eluted DNA were purified using Macherey-Nagel™ NucleoSpin™ Gel and PCR Clean-up Kit (Fisher Scientific) and subjected to quantitative PCR using PowerUp™ SYBR™ Green Master Mix (A25780, Thermo Fisher Scientific) in a ViiA7 Real-Time PCR machine (Thermo Fisher Scientific). The PCR conditions are Step 1: 95 °C for 10 min (1 cycle), and Step 2: 95 °C for 15 s, 60 °C for 1 min (40 cycles). For RNaseH digest experiments, DNA were digested with 4 μl RNaseH (M0297S, NEB) at 37 °C overnight and purified using Macherey-Nagel™ NucleoSpin™ Gel and PCR Clean-up Kit (Fisher Scientific) before subjected to DRIP as described above. Forward and reverse primers from the QAOS assay and two control loci were used for qPCR following DRIP, as listed in Supplementary Table 1.

**Chromatin immunoprecipitation (ChIP).** RPE1 cells were tripsinised, resuspended in PBS and subjected to crosslinking in 1% formaldehyde for 10 min. Glycine was added (final concentration 0.2 M) to stop the crosslinking. Cells were washed twice in PBS and resuspended in lysis buffer (50 mM Tris-HCl pH 8, 10 mM EDTA, 1% SDS, 1 mM PMSF, 1/100 protease inhibitor cocktail III Calbiochem 539134), left on ice for 10 min and sonicated for 20 min in a Diagenode Bioruptor (4 °C, high setting, 30 s on/30 s off). Supernatant were removed following centrifugation at 16,200×g for 10 min, diluted 10× in IP dilution buffer (0.01% SDS, 1.1% Triton X-100, 1.2 mM EDTA, 16.7 mM Tris-HCl pH 8.0, 150 mM NaCl, 1 mM PMSF, 1/100 protease inhibitor cocktail III Calbiochem 539134) and subjected to ChIP using anti-UPF1 goat polyclonal antibody (A300-038A, Cambridge Biosciences) or a mock antibody (goat IgG control, AB-108-C, Bio Techne Ltd) on a rotator at 4 °C overnight. Protein G dynabead (10004D, Thermo Fisher Scientific) were added the next day and the tubes were left on the rotator at 4 °C for 1 h 15 min. Dynabeads were isolated using magnet and washed successively (two times

each) with 1 ml low salt buffer (0.1% SDS, 1% Triton X-100, 2 mM EDTA pH 8.0, 20 mM Tris-HCl pH 8.0, 150 mM NaCl), 1 ml high salt buffer (0.1% SDS, 1% Triton X-100, 2 mM EDTA pH 8.0, 20 mM Tris-HCl pH 8.0, 500 mM NaCl), 1 ml wash buffer (0.25 M LiCl, 1% NP-40, 1% Na-Deoxycholate, 1 mM EDTA pH 8.0, 10 mM Tris-HCl pH 8.0), and 1 ml TE (10 mM Tris-HCl pH 8, 1 mM EDTA pH 8), before being eluted in 100 μl of elution buffer (50 mM Tris-HCl pH 8, 10 mM EDTA, 1% SDS) at 65 °C for 10 min. Eluted DNA were incubated at 65 °C overnight following addition of NaCl (0.19 M) to reverse the crosslink. Following digest with RNaseA (0.2 mg/ml at 37 °C for 30 min) and Proteinase K (0.2 mg/ml at 55 °C for 1.5 h), DNA were purified using QIAamp DNA Micro Kit (56304, Qiagen) and subjected to quantitative PCR as described in the DRIP-qPCR section.

**RNA isolation, reverse transcription, and qPCR.** RNA isolation was performed using innuPREP RNA Mini Kit (BM-845-KS-2040050, Thistle Scientific) according to manufacturer's protocol. RNAs were quantified using Nanodrop One spectrophotometer and reverse transcribed using High-Capacity cDNA Reverse Transcription Kit (4368814, Thermo Fisher Scientific). cDNAs were subjected to quantitative PCR using PowerUp™ SYBR™ Green Master Mix (A25780, Thermo Fisher Scientific) in a ViiA7 Real-Time PCR machine (Thermo Fisher Scientific). The PCR conditions are Step 1: 95 °C for 10 min (1 cycle), and Step 2: 95 °C for 15 s, 60 °C for 1 min (40 cycles). Primers used are listed in Supplementary Table 1.

**Data analysis.** The following software were used for data collection and analysis: BD FACSDiva™ 8.0.1 (BD Biosciences), NC-3000™ NucleoView v2.1 (Chemometec), ImageQuant TL v8.1 (GE Healthcare), ViiA™ 7 v1.2.3 (Thermo Fisher Scientific), QuantStudio v1.3 (Thermo Fisher Scientific), GraphPad Prism 5/9 (GraphPad Software), Flowjo v10 (BD Biosciences), Applied Biosystems qPCR Analysis Modules (Thermo Fisher Scientific), Image J v1.52 (NIH), Microsoft Excel v16 (Microsoft), and Adobe Photoshop CS v5.1 (Adobe).

**Statistical analysis.** All data generated were taken from distinct samples. Values presented represent the means + standard error of the mean (SEM). The number of repeats ($n$) are indicated in the figure legends. $p$ values were obtained using Student's $t$-test (unpaired, equal variance, one-tailed or two-tailed as indicated in the figure legends) or two-tailed Mann–Whitney $T$-test (for analysis of telomere length data).

**Reporting summary.** Further information on research design is available in the Nature Research Reporting Summary linked to this article.

## Data availability
All data generated in this study are included in this article. Source data are provided with this paper.

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

## Acknowledgements

This work was funded by Cancer Research UK (A18246/A29202). The authors would like to thank Prof. Eric Hendrickson (University of Minnesota, USA) for providing the RPE1-hTERT WT and HCT116 WT, *LIG3* KO, and *LIG4* KO cell lines, Prof. Jeremy Stark (City of Hope Comprehensive Cancer Centre, USA) for providing the U2OS EJ5-GFP, DR-GFP, and SA-GFP reporter cell lines.

## Author contributions

G.N. designed experiments, performed experiments, analysed data and wrote the manuscript; J.W.G. assisted with experimental procedures; D.B. supervised the study and edited the manuscript.

## Competing interests

The authors declare no competing interests.
