## [Peer Review File · Nature Communications]

REVIEWER COMMENTS

Reviewer #1 (Remarks to the Author):

Ngo et al. investigate repair of TALEN-generated ds breaks in human subtelomeres, using their single-molecule telomere fusion assay to analyze repair products and a quantitative amplification of single-stranded DNA assay (QAOS) to measure specific single-stranded DNAs generated by break processing. The most important findings are that R-loops are formed at subtelomeric ds breaks independently of DNA resection and occur alongside 2-stranded RNA/DNA hybrids that form on ssDNA created by resection. Further, they show that UPF1 is a critical factor that drives formation of R-loops and RNA/DNA hybrids, stimulating DNA resection and repair of ds breaks. The UPF1 functions in DNA repair appear to be independent of its role in nonsense-mediated decay. While UPF1 is especially interesting in the context of its previously discovered telomere localization, interaction with the telomeric noncoding RNA TERRA, and potential role in telomere stability and function, it is also shown to stimulate repair of non-telomeric DNA damage.

The results include detection of 5' ssDNA repair intermediates that are sensitive to RNaseH1 using QAOS at subtelomeric DNA breaks, direct evidence of R-loops generated at ds breaks. This, and the convincing demonstration of UPF1 involvement in stimulating DNA repair activities at subtelomeres as well as genome-wide, are important advances in our understanding of DNA damage response and repair activities; in addition, they lead to intriguing questions as to the functions of TERRA in the context of its interactions with UPF1 and TERRA's potential to form R-loops with telomeric DNA in a variety of situations with functional consequences.

The paper is written clearly and logically, with the experimental results supporting the conclusions. The innovative methodology seems sound, with interesting adaptations of single-molecule fusion-PCR methods to measure the relative contribution of end-joining pathways and gather evidence for levels of DNA resection prior to end-joining. Adaptation of QAOS assays developed in yeast to human genomes is also novel, with detection sensitivities presumably enhanced by the number of subtelomere loci with nearly-identical subtelomere breaks induced by the TALEN experimental system. Multiple orthogonal validation experiments buttressed the key conclusions, which was a strength of the paper.

A couple of minor points to comment upon:

One of the UPF1 knockout lines (46) appears to retain low levels of UPF1 and has somewhat different phenotypic properties compared to the other where no protein is detected. What is the evidence that 46 is a true knockout ?

The processing of the DNA on the telomeric side of the subtelomeric ds break appears to be significantly different than on the centromeric side (figs 5 and 6), with higher levels of RNA/DNA hybrids and possibly R-loops on the telomeric side. How might telomeric chromatin/shelterin interactions and TERRA binding affect the interpretation of these results ?

Reviewer #2 (Remarks to the Author):

The manuscript by Duncan et al. is a well written and interesting paper describing for the first time the formation of R-loops near DSB sites in a pathway that parallels the previously described formation of DNA-RNA hybrids. This is observed at subtelomeric DSB and it would be interesting to know whether it also apply to intrachromosomal DSB. In addition, the author found that the formation of both R-loops and DNA-RNA hybrids rely on the RNA helicase UPF1 which is recruited to DSB sites. They convincingly show that this occurs independently of UPF1 activity in the NMD pathway depicting a new function for UPF1. Their experiments convincingly show that UPF1 dependent DNA-RNA hybrid formation and that UPF1 contribute to DSB resection and suggest that this regulates DSB repair. Overall, this story is very interesting, this manuscript is well written, the data are of good quality and it should be of interest for a broad audience.

Detailed concerns:

Figure 1C : It would be informative to add the original 21q and 16p sequences and the cut site to help the reader to understand the repair event (no microhomology).

Figure 1J: The effect of mirin treatment is not striking from the blot shown. MRN, thus seems marginally required to generate large deletion. Since other pathways, like Exo1, are also participating in DSB-end resection but are not inhibited by mirin, it is far fetched to claim that resection is responsible for the large deletions observed at telomere fusions. The authors should be more cautious with their conclusion on that point and could test whether the inhibition of several resection pathways enhances the tiny effect seen with Mirin.

Figure 2: The authors convincingly show that the NMD defect caused by UPF1 KO are complemented by transient expression of UPF1. They should also show whether UPF1 KO complementation also suppresses telomere fusions. Especially when considering the difference they observe between UPF1 KD and UPF1 KO. To explain the differences between these two experiments they suggest off target effects in UPF1 KD. Alternatively, there could also be suppressors accumulated in UPF1 KO. The Authors should be more cautious on their interpretation.

Figure 3A: I have a conceptual problem with this result that shows less 3' ssDNA formed at 0.5 kb than at 1.2 kb. How is this possible? In the following panels ssDNA is always more abundant at 0.5kb than at 1.2kb. The same problem is found in figure 6A.

Figure 3I: The effect observed in UPF3 KO on resection are relatively minor (less than 2 folds = not 'strong') compared to UPF1 KO (3 folds). The authors should minor their assertion on the strong effect and the fact that it is comparable to UPF1 although going in the same direction.

Figure 4E, F, G and S4: to make these test accessible to a wide audience, I suggest to add schemes depicting the different assays. What is the assay used for HR?

To strengthen their conclusion could the authors use an assay in which DSB repair occurs independently of resection and in which UPF1 KD should have no effect?

Figure 5C-E: Why do the authors show only telomere 17p as an uncut control? They should also assess DRIP at 21p in untransformed cells.

Reviewer #3 (Remarks to the Author):

This manuscript shows that the RNA helicase UPF1 promotes DNA double-strand break (DSB) repair by stimulating the formation of both two-stranded DNA-RNA hybrids and three-stranded R loops in human cells. The experimental system is based on the formation of gene fusions between the 21q and 16p telomeres using TALEN. Prior to the identification of UPF1 as a player in telomere fusion, DNA resection mediated by Mre11 is shown to promote telomere fusion by generating deletion at sub-telomeric DSBs, which is followed by either nonhomologous end-joining or microhomology-mediated end-joining. Then, UPF1, identified by siRNA screening, is shown to promote telomere deletion by stimulating DNA resection independently of its nonsense-mediated mRNA decay activity. In this part, in order to quantify the level of 3' ssDNA at DSBs, the quantitative amplification of ss DNA (QAOS) analysis was employed. Next, apart from its specific role at sub-telomeric DSBs, UPF1's general roles in checkpoint activation, single-strand annealing, homologous recombination and end-joining are demonstrated. Based on these findings, the roles of UPF1 in DNA-RNA hybrid formation as well as R loop formation were investigated, and both roles are demonstrated. In the last part, results revealed by the QAOS analysis designed to quantify 5' ssDNA as well as 3' ssDNA strengthen the role of UPF1 in the formation of R loops.

These findings provide novel insight into the role of UPF1 in the maintenance of genome integrity. However, evidence suggesting its general role in DNA damage response is weak. In Fig 4A and 4B, levels of phosphorylated RPA, CHK1 and CHK2 are demonstrated. In order to confirm these results, their levels of total proteins including non-phosphorylated forms should be shown in parallel with those of phosphorylated forms. Given that activating roles of UPF1 in the DNA damage checkpoint pathways are confirmed, it is interesting to know whether upstream regulators such as ATM are affected by UPF1. This can be investigated by immunofluorescence or western blots.

Minor points

1. A method of an siRNA screen to examine how the silencing of a panel of DNA damage response genes affects the formation of deletions should be briefly described.
2. P12, L5. IScel is I-SceI.
3. P12, L7. Is there direct evidence that checkpoint activation is DNA resection-dependent?

REVIEWERS' COMMENTS

Reviewer #1 (Remarks to the Author):

Ngo et al. investigate repair of TALEN-generated ds breaks in human subtelomeres, using their single-molecule telomere fusion assay to analyze repair products and a quantitative amplification of single-stranded DNA assay (QAOS) to measure specific single-stranded DNAs generated by break processing. The most important findings are that R-loops are formed at subtelomeric ds breaks independently of DNA resection and occur alongside 2-stranded RNA/DNA hybrids that form on ssDNA created by resection. Further, they show that UPF1 is a critical factor that drives formation of R-loops and RNA/DNA hybrids, stimulating DNA resection and repair of ds breaks. The UPF1 functions in DNA repair appear to be independent of its role in nonsense-mediated decay. While UPF1 is especially interesting in the context of its previously discovered telomere localization, interaction with the telomeric noncoding RNA TERRA, and potential role in telomere stability and function, it is also shown to stimulate repair of non-telomeric DNA damage.

The results include detection of 5' ssDNA repair intermediates that are sensitive to RNaseHI using QAOS at subtelomeric DNA breaks, direct evidence of R-loops generated at ds breaks. This, and the convincing demonstration of UPF1 involvement in stimulating DNA repair activities at subtelomeres as well as genome-wide, are important advances in our understanding of DNA damage response and repair activities; in addition, they lead to intriguing questions as to the functions of TERRA in the context of its interactions with UPF1 and TERRA's potential to form R-loops with telomeric DNA in a variety of situations with functional consequences.

The paper is written clearly and logically, with the experimental results supporting the conclusions. The innovative methodology seems sound, with interesting adaptations of single-molecule fusion-PCR methods to measure the relative contribution of end-joining pathways and gather evidence for levels of DNA resection prior to end-joining. Adaptation of QAOS assays developed in yeast to human genomes is also novel, with detection sensitivities presumably enhanced by the number of subtelomere loci with nearly-identical subtelomere breaks induced by the TALEN experimental system. Multiple orthogonal validation experiments buttressed the key conclusions, which was a strength of the paper.

We thank the reviewer for their supportive comments on the manuscript.

A couple of minor points to comment upon:

One of the UPF1 knockout lines (46) appears to retain low levels of UPF1 and has somewhat different phenotypic properties compared to the other where no protein is detected. What is the evidence that 46 is a true knockout ?

We agree that UPF1 46 is not a complete knockout and have modified the sentence below, page 7 of the revised manuscript.

To confirm this finding, we used CRISPR-Cas9D10A nickase²⁶ to knockout (KO) UPF1 in RPE1 cells and isolated two clones (46 and 54) which showed severe depletion of the UPF1 protein, with UPF1 54 being a more complete KO than UPF1 46, which is a partial KO (Figure 2c).

The processing of the DNA on the telomeric side of the subtelomeric ds break appears to be significantly different than on the centromeric side (figs 5 and 6), with higher levels of RNA/DNA hybrids and possibly R-loops on the telomeric side. How might telomeric chromatin/shelterin interactions and TERRA binding affect the interpretation of these results ?

We thank the reviewer for raising this interesting question. We believe that some of the RNA/DNA hybrids detected near these breaks are indeed the result of TERRA binding to the exposed DNA. The higher level of RNA/DNA hybrids observed on the telomeric side of the DSBs could be explained by the strand specificity of TERRA, which are mainly transcribed from the 'bottom' C rich strand towards the telomeres, and therefore could only bind to 3' ssDNA on the telomeric side, but not to 3' ssDNA on the centromeric side of these DSBs, thus only increasing the level of RNA/DNA hybrids on the telomeric side. However as our experiments did not directly address this issue we have refrained from speculation about the role of TERRA.

Reviewer #2 (Remarks to the Author):

The manuscript by Duncan et al. is a well written and interesting paper describing for the first time the formation of R-loops near DSB sites in a pathway that parallels the previously described formation of DNA-RNA hybrids. This is observed at subtelomeric DSB and it would be interesting to know whether it also apply to intrachromosomal DSB. In addition, the author found that the formation of both R-loops and DNA-RNA hybrids rely on the RNA helicase UPF1 which is recruited to DSB sites. They convincingly show that this occurs independently of UPF1 activity in the NMD pathway depicting a new function for UPF1. Their experiments convincingly show that UPF1 dependent DNA-RNA hybrid formation and that UPF1 contribute to DSB resection and suggest that this regulates DSB repair.

Overall, this story is very interesting, this manuscript is well written, the data are of good quality and it should be of interest for a broad audience.

We thank the reviewer for these encouraging comments.

We agree that it would be interesting to examine the formation of R loops at intrachromosomal DSB, but we found that it is more challenging for using QAOS to detect strand specific ssDNA at single copy loci. Thus, it will require a lot of effort to further optimise the assay, and to find specific loci that are prone to form R loops. Therefore, we consider that this is outside the scope of this manuscript.

Detailed concerns:

Figure 1C : It would be informative to add the original 21q and 16p sequences and the cut site to help the reader to understand the repair event (no microhomology).

We have now added these sequences (Supplementary Fig. 1a) and realised that the level of deletion shown in Fig. 1c,f,g should be 2 nucleotide longer, we have now corrected this error and modified the sentences below on page 5.

The fused 16p telomere was deleted by 59bp whereas the fused 21q telomere was deleted by 2bp and there was no microhomology at the fusion junction, implicating classical-non homologous end joining (c-NHEJ) in the fusion (Figure 1c).

We have also adjusted the distance of the primers and telomeres to the cleavage site accordingly on page 5.

The 21q/16p TALEN cleaves the 21q and 16p family telomeres at a site which is located at 1494bp (21q) or 1027bp (16p) from the start of the telomere repeat arrays⁴

Primer 16p1 recognises a sequence located at 5058bp proximal to the cleavage site on the 16p telomeres, whereas primer 21q1 recognises a sequence located at either 2176bp or 1856bp proximal from the cleavage site.

Figure 1J: The effect of mirin treatment is not striking from the blot shown. MRN, thus seems marginally required to generate large deletion. Since other pathways, like Exo1, are also participating in DSB-end resection but are not inhibited by mirin, it is far fetch to claim that resection is responsible for the large deletions observed at telomere fusions. The authors should be more cautious with their conclusion on that point and could test whether the inhibition of several resection pathways enhances the tiny effect seen with Mirin.

We thank the reviewer for this suggestion. We have now included new data showing that simultaneous inhibition of Mre11 and BLM strongly reduces the level of deletion at these DSBs (Supplementary Fig 1h,1i), which support the idea that resection is responsible for generating these large deletion events. This result could reflect the overlapping role of MRN and BLM in stimulating both EXO1 and DNA2 (Nimonkar et.al 2011) and support a previous finding that co-inhibition of these proteins strongly reduce deletion at DSBs (Grabarz et.al 2013). The text on page 7 has been amended as follows:

In addition, co-inhibition of Mre11 and BLM, which plays overlapping role in activating DNA resection²⁶, further reduced the level of deletion (Supplementary Fig.1h,1i).

Nimonkar, A.V., Genschel, J., Kinoshita, E., Polaczek, P., Campbell, J.L., Wyman, C., Modrich, P., and Kowalczykowski, S.C. (2011). BLM-DNA2-RPA-MRN and EXO1-BLM-RPA-MRN constitute two DNA end resection machineries for human DNA break repair. *Genes Dev* 25, 350-362.

Grabarz, A., Guirouilh-Barbat, J., Barascu, A., Pennarun, G., Genet, D., Rass, E., Germann, S.M., Bertrand, P., Hickson, I.D., and Lopez, B.S. (2013). A role for BLM in double-strand break repair pathway choice: prevention of CtIP/Mre11-mediated alternative nonhomologous end-joining. *Cell reports* 5, 21-28.

Figure 2: The authors convincingly show that the NMD defect caused by UPF1 KO are complemented by transient expression of UPF1. They should also show whether UPF1 KO complementation also suppresses telomere fusions.

We have now included data examining the effect of transient UPF1 expression on telomere fusions (Supplementary Fig.2 e-g).

Intriguingly, we found that transient expression of UPF1 reduced deletion further in UPF1 KO cells, in addition to reducing insertion, suggesting a dominant negative effect, which is confirmed in WT cells (Supplementary Fig.2 e-g).

Consistent with this observation, previous literature has documented that overexpression of proteins can also lead to similar phenotypes as observed with loss of function mutations of the same proteins (Prelich 2012). This is commonly observed when the protein works within a multiprotein complex, with overexpression disrupting the stoichiometry or interfering with the function of the complex (Prelich 2012).

We propose that the role of UPF1 in DNA repair, unlike its role in cytoplasmic NMD, is highly sensitive to changes in its level, and ectopic expression of UPF1 could produce a dominant negative effect by disrupting a UPF1 complex within the nucleus. In support of this, this dominant negative effect of UPF1 overexpression seems to be influenced by the level of the endogenous UPF1, as it is stronger in WT and UPF1 46, and weaker in UPF1 54, which has the least amount of endogenous UPF1.

We have added the following sentences on pages 8-9 in the revised manuscript to explain this finding.

Intriguingly, transient expression of UPF1 did not restore, but further reduced the level of deletion, in addition to reducing insertion events in UPF1 KO cells, with the effect being stronger in UPF1 46 than UPF1 54 (Supplementary Fig.2 e-f), suggesting that ectopic expression of UPF1, could exert a dominant negative effect on telomere processing, despite having a positive effect on NMD (Supplementary Fig.2b). This dominant negative phenotype was also observed in WT cells (Supplementary Fig.2g). This result suggests that the function of UPF1 at telomere repair could be distinct from its role in NMD and that this function is highly sensitive to perturbation in the level of UPF1. We conclude that UPF1 strongly stimulates deletion formation and could also affect insertion at sub-telomeric DSBs.

Prelich, G. (2012). Gene overexpression: uses, mechanisms, and interpretation. *Genetics* 190, 841-854.

Especially when considering the difference they observe between UPF1 KD and UPF1 KO. To explain the differences between these two experiments they suggest off target effects in UPF1 KD. Alternatively, there could also be suppressors accumulated in UPF1 KO. The Authors should be more cautious on their interpretation.

We agree with the reviewer that we need to be more cautious with this interpretation.

Our data suggest that UPF1 KD has an additional phenotype of reducing insertion events. This could be a genuine phenotype, given the observation above that transient expression of UPF1 also mirror this phenotype by reducing both deletion and insertion.

Thus, it is possible that the role of UPF1 on insertion is less obvious in the KO due to suppressor accumulation or alternatively genetic compensation.

We have included the following sentence on page 8 of the revised manuscript.

Alternatively, the effect on insertion could be masked by suppressor accumulation or genetic compensation in UPF1 KO cells.

Figure 3A: I have a conceptual problem with this result that shows less 3' ssDNA formed at 0.5 kb than at 1.2 kb. How is this possible ? In the following panels ssDNA is always more abundant at 0.5kb than at 1.2kb. The same problem is found in figure 6A.

We believe that one possible explanation for this observation is that 3' ssDNA generated earlier (for example at 0.5kb) could be subjected to longer period of degradation or annealing with DNA or RNA, thus reducing the level of these 3' ssDNA. We have previously observed this trend at some loci in our analysis of DNA resection in budding yeast (Ngo et.al 2014).

The question as to why 5' ssDNA (Figure 6a) is more abundant at 1.2kb than 0.5kb is more complicated and we speculate this may involve different propensity of these regions to support the formation of stable R loops.

Ngo, G.H., Balakrishnan, L., Dubarry, M., Campbell, J.L., and Lydall, D. (2014). The 9-1-1 checkpoint clamp stimulates DNA resection by Dna2-Sgs1 and Exo1. *Nucleic Acids Res* 42, 10516-10528.

Figure 3I: The effect observed in UPF3 KO on resection are relatively minor (less than 2 folds = not 'strong') compared to UPF1 KO (3 folds). The authors should minor their assertion on the strong effect and the fact that it is comparable to UPF1 although going in the same direction.

We agree with the reviewer and have toned this point down by modifying the sentences below on page 11 of the revised manuscript:

Interestingly, these cells exhibited a modest NMD defect compared to UPF1 KO cells (Figures 3h and 2f) but nevertheless had a similar defect in stimulating DNA resection which was comparable to the defect observed in UPF1 KO 46 (Figures 3i and 3d), again suggesting that the defects in DNA resection observed in UPF1 and UPF3B KO is not linked to reductions in NMD.

Figure 4E, F, G and S4: to make these test accessible to a wide audience, I suggest to add schemes depicting the different assays. What is the assay used for HR?

We have now added the schematics as suggested (Figures 4e-h). The HR assay used is an intra-chromosomal gene conversion assay as depicted in Fig 4f.

To strengthen their conclusion could the authors use an assay in which DSB repair occurs independently of resection and in which UPF1 KD should have no effect?

We have now included an NHEJ assay which allows minimal resection and showed that UPF1 does not affect this process (new Figure 4h).

We have added the following sentence on page 13 of the revised manuscript:

We also examined the role of UPF1 in C-NHEJ by using a plasmid recircularization assay (Figure 4h), which is highly dependent on LIG4 (Supplementary Fig. 4d) and showed that UPF1 plays no role in this repair pathway (Figure 4h).

Figure 5C-E: Why do the authors show only telomere 17p as an uncut control ? They should also assess DRIP at 21p in untransformed cells.

We chose 17p because the sequence has been well characterised in our lab. We have now included DRIP analysis at a locus located at ~ 2kb from 21p telomere and showed that TALEN did not induce the formation of DNA-RNA hybrid at this locus (Fig. 5c)

We have modified the following sentence on page 14 of the revised manuscript.

In support of the observed induction of DNA-RNA hybrids at DSBs¹⁰⁻¹³, we detected increased level of DNA-RNA hybrids on both sides of the DSBs (0.5kb, 1.2kb, 3.5kb on centromeric side and 1.1kb on telomeric side) but not at an uncut 17p or 21p telomere (Figure 5c).

Reviewer #3 (Remarks to the Author):

This manuscript shows that the RNA helicase UPF1 promotes DNA double-strand break (DSB) repair by stimulating the formation of both two-stranded DNA-RNA hybrids and three-stranded R loops in human cells. The experimental system is based on the formation of gene fusions between the 21q and 16p telomeres using TALEN. Prior to the identification of UPF1 as a player in telomere fusion, DNA resection mediated by Mre11 is shown to promote telomere fusion by generating deletion at sub-telomeric DSBs, which is followed by either nonhomologous end-joining or microhomology-mediated end-joining. Then, UPF1, identified by siRNA screening, is shown to promote telomere deletion by stimulating DNA resection independently of its nonsense-mediated mRNA decay activity. In this part, in order to quantify the level of 3' ssDNA at DSBs, the quantitative amplification of ss DNA (QAOS) analysis was employed. Next, apart from its specific role at sub-telomeric DSBs, UPF1's general roles in checkpoint activation, single-strand annealing, homologous recombination and end-joining are demonstrated. Based on these findings, the roles of UPF1 in DNA-RNA hybrid formation as well as R loop formation were investigated, and both roles are demonstrated. In the last part, results revealed by the QAOS analysis designed to quantify 5' ssDNA as well as 3' ssDNA strengthen the role of UPF1 in the formation of R loops.

These findings provide novel insight into the role of UPF1 in the maintenance of genome integrity. However, evidence suggesting its general role in DNA damage response is weak. In Fig 4A and 4B, levels of phosphorylated RPA, CHK1 and CHK2 are demonstrated. In order to confirm these results, their levels of total proteins including non-phosphorylated forms should be shown in parallel with those of phosphorylated forms. Given that activating roles of UPF1 in the DNA damage checkpoint pathways are confirmed, it is interesting to know whether upstream regulators such as ATM are affected by UPF1. This can be investigated by immunofluorescence or western blots.

We thank the reviewer for this suggestion which make the analysis more accurate. We have replaced the data with new western blots showing the level of total proteins, included ATM-S1981p in our analysis (Figure 4a), and normalised the level of phosphoproteins with the level of total proteins (Figure 4b). We found that UPF1 KO cells had a stronger defect in the phosphorylation of RPA32 and CHK1, compared to ATM or CHK2. We have modified the following sentences on page 12 of the revised manuscript.

To assess whether UPF1 regulates DNA repair at non-telomeric DSBs, we treated RPE1 WT and UPF1 KO cells with bleomycin and examined the phosphorylation state of DNA damage response proteins after 90 minutes. Both UPF1 KO clones had a strong defect in the phosphorylation of RPA32 and CHK1, but had a weaker effect on the phosphorylation of ATM and its downstream target CHK2 (Figures 4a and 4b). This observation suggests that UPF1 may also stimulate DNA resection at non-telomeric DSBs, as resection has been shown to be required for the sustained activation of RPA32 and CHK1 to maintain the activation of the DNA damage checkpoint³⁸.

Minor points

1. A method of an siRNA screen to examine how the silencing of a panel of DNA damage response genes affects the formation of deletions should be briefly described.

We have now included this in the method section on page 25 of the revised manuscript as detailed below:

siRNA screen

1.9×10^5 RPE1 cells were plated into 12 well plates on day 1. The cells were transfected on day 2 with 50nM siRNA (Dharmacon siGENOME[®] SMARTpool[®] siRNA Library targeting Human DNA Damage Response, G-006005 Lot 14019, divided into 12 sets) using 2.5 μ l Dharmafect 4 (Dharmacon) according to manufacturer's protocol. 4×10^5 siRNA transfected cells were passed onto 6 well plates on day 3 and transfected with 21q TALEN plasmids on day 4 using 3.75 μ l Dharmafect kb (Dharmacon) according to manufacturer's protocol. Cells were harvested at day 7 and DNA were purified before subjected to fusion analysis.

2. P12, L5. ISceI is I-SceI.

We have corrected this typographical error in the revised manuscript.

3. P12, L7. Is there direct evidence that checkpoint activation is DNA resection-dependent?

We thank the reviewer for raising this point.

It has indeed been shown that DNA resection is not required for checkpoint initiation, but it is essential for the maintenance of checkpoint activity (Kousholt et.al 2012).

We have modified the following sentences below on page 12 to better clarify this point.

We next examined bleomycin induced DNA damage checkpoint maintenance in these cells.

We conclude that UPF1 stimulates the maintenance of DNA damage checkpoint following non-telomeric DSB induction.

Kousholt, A.N., Fugger, K., Hoffmann, S., Larsen, B.D., Menzel, T., Sartori, A.A., and Sorensen, C.S. (2012). CtIP-dependent DNA resection is required for DNA damage checkpoint maintenance but not initiation. *J Cell Biol* 197, 869-876.

REVIEWERS' COMMENTS

Reviewer #1 (Remarks to the Author):

In this revised version of the manuscript "UPF1 promotes the formation of R loops to stimulate DNA double-strand break repair" the authors provide detailed constructive responses to each of the reviewer comments, including additional data to support their responses to the comments and criticisms. In my view they have adequately addressed the reviewer comments, and made an already well-written and important paper even better.

Reviewer #2 (Remarks to the Author):

The authors have adequately addressed my concern and should be considered for publication.

Reviewer #3 (Remarks to the Author):

In the revised manuscript, the authors mainly focus on four additional experiments. First, to confirm the role of DNA resection in sub-telomeric deletion, the effect of BLM inhibition was investigated. Consequently, co-inhibition of MRE11 and BLM reduced the deletion level to approximately 10% of the control, indicating that MRE11-dependent resection contributes to deletion. Second, to confirm the role of UPF1 in sub-telomeric deletion, the effect of its transient expression in UPF1 KO cells was investigated. Unexpectedly, UPF1 expression further reduced the deletion levels both in KO cells and in wild-type cells. This result can be explained by a dominant-negative effect of UPF1 exogenous overexpression on deletion though the complementing phenotype was not observed. Third, the role of UPF1 in the G2 checkpoint was confirmed by additional western blots. Lastly, no role of UPF1 in C- NHEJ was confirmed by using a plasmid ligation assay. Overall, these additional experiments strengthen the original conclusion.

REVIEWERS' COMMENTS

Reviewer #1 (Remarks to the Author):

In this revised version of the manuscript "UPF1 promotes the formation of R loops to stimulate DNA double-strand break repair" the authors provide detailed constructive responses to each of the reviewer comments, including additional data to support their responses to the comments and criticisms. In my view they have adequately addressed the reviewer comments, and made an already well-written and important paper even better.

We thank the reviewer for these encouraging comments.

Reviewer #2 (Remarks to the Author):

The authors have adequately addressed my concern and should be considered for publication.

We thank the reviewer for the positive response.

Reviewer #3 (Remarks to the Author):

In the revised manuscript, the authors mainly focus on four additional experiments. First, to confirm the role of DNA resection in sub-telomeric deletion, the effect of BLM inhibition was investigated. Consequently, co-inhibition of MRE11 and BLM reduced the deletion level to approximately 10% of the control, indicating that MRE11-dependent resection contributes to deletion. Second, to confirm the role of UPF1 in sub-telomeric deletion, the effect of its transient expression in UPF1 KO cells was investigated. Unexpectedly, UPF1 expression further reduced the deletion levels both in KO cells and in wild-type cells. This result can be explained by a dominant-negative effect of UPF1 exogenous overexpression on deletion though the complementing phenotype was not observed. Third, the role of UPF1 in the G2 checkpoint was confirmed by additional western blots. Lastly, no role of UPF1 in C- NHEJ was confirmed by using a plasmid ligation assay. Overall, these additional experiments strengthen the original conclusion.

We thank the reviewer for these supportive comments.